# High-phytate/low-calcium diet is a risk factor for crystal nephropathies, renal phosphate wasting, and bone loss

Ok-Hee Kim[1], Carmen J Booth[2], Han Seok Choi[3], Jinwook Lee[1], Jinku Kang[1], June Hur[1], Woo Jin Jung[1], Yun-Shin Jung[1], Hyung Jin Choi[4], Hyeonjin Kim[5], Joong-Hyuck Auh[6], Jung-Wan Kim[7], Ji-Young Cha[8], Young Jae Lee[8], Cheol Soon Lee[9], Cheolsoo Choi[10], Yun Jae Jung[11], Jun-Young Yang[12], Seung-Soon Im[13], Dae Ho Lee[10], Sun Wook Cho[14], Young-Bum Kim[15], Kyong Soo Park[14], Young Joo Park[14], Byung-Chul Oh[1]*

[1]Department of Physiology, Lee Gil Ya Cancer and Diabetes Institute, Gachon University College of Medicine, Incheon, Republic of Korea; [2]Department of Comparative Medicine, Yale University School of Medicine, New Haven, United States; [3]Division of Endocrinology and Metabolism, Department of Internal Medicine, Dongguk University Ilsan Hospital, Goyang, Republic of Korea; [4]Department of Anatomy, Seoul National University College of Medicine, Seoul, Republic of Korea; [5]Department of Radiology, Seoul National University College of Medicine, Seoul, Republic of Korea; [6]Department of Food Science and Technology, Chung-Ang University, Ansung, Republic of Korea; [7]Department of Biology, University of Incheon, Incheon, Republic of Korea; [8]Department of Biochemistry, Lee Gil Ya Cancer and Diabetes Institute Gachon University College of Medicine, Incheon, Republic of Korea; [9]Medical Health Research Center, Korea University Anam Hospital, Seoul, Republic of Korea; [10]Department of Internal Medicine, Gachon University Gil Medical Center, Incheon, Republic of Korea; [11]Department of Mirobiolgy, Lee Gil Ya Cancer and Diabetes Institute, Gachon University College of Medicine, Incheon, Republic of Korea; [12]Department of Toxicological Evaluation and Research, Ministry of Food and Drug Safety, Cheongju-si, Republic of Korea; [13]Department of Physiology, Keimyung University School of Medicine, Daegu, Republic of Korea; [14]Department of Internal Medicine, Seoul National University College of Medicine, Seoul, Republic of Korea; [15]Division of Endocrinology, Diabetes, and Metabolism, Beth Israel Deaconess Medical Center and Harvard Medical School, Boston, United States

*For correspondence:
bcoh@gachon.ac.kr

Competing interests: The authors declare that no competing interests exist.

**Abstract** Phosphate overload contributes to mineral bone disorders that are associated with crystal nephropathies. Phytate, the major form of phosphorus in plant seeds, is known as an indigestible and of negligible nutritional value in humans. However, the mechanism and adverse effects of high-phytate intake on $Ca^{2+}$ and phosphate absorption and homeostasis are unknown. Here, we show that excessive intake of phytate along with a low-$Ca^{2+}$ diet fed to rats contributed to the development of crystal nephropathies, renal phosphate wasting, and bone loss through tubular dysfunction secondary to dysregulation of intestinal calcium and phosphate absorption. Moreover, $Ca^{2+}$ supplementation alleviated the detrimental effects of excess dietary phytate on bone and kidney through excretion of undigested $Ca^{2+}$-phytate, which prevented a vicious cycle of intestinal phosphate overload and renal phosphate wasting while improving intestinal $Ca^{2+}$ bioavailability. Thus, we demonstrate that phytate is digestible without a high-$Ca^{2+}$ diet and is a

risk factor for phosphate overloading and for the development of crystal nephropathies and bone disease.

## Introduction

Phytate (*Myo*-inositol hexaphosphate) is the main phosphorus storage system in plant seeds such as those of cereal grains, nuts, oilseeds, and legumes (*Raboy, 2000*; *Reddy et al., 1982*), accounting for up to 80% of total phosphorous. Phytate tightly binds to essential minerals such as $Ca^{2+}$, $Mg^{2+}$, $Fe^{2+}$, and $Zn^{2+}$ to form mineral-phytate salts because of its six phosphate groups (*Kim et al., 2010*; *Oh et al., 2006*). These mineral-phytate salts are known to be indigestible by monogastric animals and humans, which lack the digestive enzyme phytase that releases phosphate from mineral-phytate salts (*Reddy and Sathe, 2002*). The prevailing hypothesis argues that phytate-derived phosphorus is indigestible, not absorbed in the intestine, and is excreted in the feces (*Golovan et al., 2001*). This widespread belief led to the development of low-phytate crops and commercial phytases to improve the bioavailability of phytate-bound essential minerals, including phosphorus, and to improve the nutritional quality of plant seeds (*Raboy, 2007*). Thus, plant seeds or high-fiber vegetable proteins represent a poor or negligible source of bioavailable phosphate for intestinal absorption compared with animal proteins (*Moe et al., 2011*; *Vervloet et al., 2017*). Further, they are not considered to affect intestinal phosphate absorption adversely in humans eating diets that are mainly comprised of plant-based, cereal-rich diets (*Reddy and Sathe, 2002*). Nonetheless, there are insufficient data indicating the amount of phytate phosphorus that is bioavailable for intestinal absorption and whether specific conditions can modulate the bioavailability and digestibility of phytate.

High-phytate diets cause stronger adverse effects on $Ca^{2+}$, $Fe^{2+}$, and $Zn^{2+}$ levels in infants, young children, and pregnant and lactating women when cereal-based foods represent a large percentage of the diet (*Al Hasan et al., 2016*; *Chan et al., 2007*). Prior studies of dogs demonstrate that increasing the levels of phytate reduces the intestinal absorption of $Ca^{2+}$ and causes nutritional rickets (*Harrison and Mellanby, 1939*; *Mellanby, 1949*). Likewise, increasing dietary phytate is associated with secondary hyperparathyroidism in young adults because of the reduced intestinal absorption of dietary $Ca^{2+}$ (*Calvo et al., 1988*; *Pettifor, 1994*). Consistent with previous observations (*Ford et al., 1972*; *Goswami et al., 2000*), a subsequent case-control study found that nutritional rickets is more prevalent in children who consume diets comprising low dietary $Ca^{2+}$ and high phytate (*Aggarwal et al., 2012*). These findings suggest a strong association between high-phytate intake and metabolic bone diseases, including nutritional rickets and osteomalacia. However, although the roles of phytate in mineral deficiency are well-defined (*Raboy, 2000*; *Reddy et al., 1982*), who do not know whether dietary phytate can modulate the intestinal absorption of phosphate because of a lack of awareness.

Despite its adverse effects on animal and human health, dietary phytate can be beneficial because it inhibits digestive enzymes that hydrolyze lipids, proteins, and starch (*Kumar et al., 2010*). For example, high-phytate diets inhibit the absorption of glucose and lipids, leading to lower serum levels of glucose, triglycerides, and total cholesterol in humans (*Thompson et al., 1987*) and rodents (*Kim et al., 2010*; *Jariwalla et al., 1990*; *Onomi et al., 2004*). Moreover, diets based on vegetable protein lead to lower plasma levels of phosphate and fibroblast growth factor 23 (FGF23), a major regulator of phosphate homeostasis (*Razzaque, 2009*), compared with those based on meat protein (*Moe et al., 2011*). Indeed, few studies suggest that phytate diets reduce the risk of developing $Ca^{2+}$-induced kidney stones (*Goswami et al., 2000*). Unfortunately, these studies fail to contribute a broader understanding of the detrimental effects of excess phytate intake on the functions of bone and kidney, two major organs that regulate $Ca^{2+}$ and phosphate homeostasis.

Here, we investigated the effects of excessive intake of phytate on $Ca^{2+}$ and phosphate homeostasis in a rat model, as well as in a dietary study of humans. Thus, the goals of this study were to determine how a high-phytate diet affects kidney and bone health, to establish whether these changes depend on $Ca^{2+}$ or mineral content, and to provide a rationale for restricting excessive phytate intake in patients who have kidney and metabolic bone diseases.

# Results

## Phytate-fed rats develop secondary hyperparathyroidism with hypophosphatemia and azotemia

We previously reported that phytate forms insoluble tricalcium- or tetracalcium-phytate salts in vitro over a wide pH range (*Kim et al., 2010*). To evaluate the systemic effects of dietary phytate, rats (4-week-old male and female Sprague Dawley) were fed AIN-93G diets supplemented with 0% (control), 1%, 3%, or 5% phytate for 12 weeks (*Figure 1A–Supplementary file 1*). On the basis of the molecular weights of $Ca^{2+}$ and phytate in these diets, increased phytate supplementation at a constant $Ca^{2+}$ concentration (0.5%) gradually decreases the molar ratio of $Ca^{2+}$/phytate (*Figure 1B*). Rats that were fed higher levels of phytate (3% and 5%) exhibited decreased growth rates and reduced fat-free lean mass and whole-body fat compared with those of the control, despite similar daily food intake (*Figure 1C–F*). We reasoned therefore that the decreased growth rates were caused by decreased bioavailability of dietary $Ca^{2+}$ associated with increased phytate supplementation.

To address this possibility, we used inductively coupled plasma mass spectroscopy (ICP-MS) to analyze fecal minerals. We found that dietary phytate led to the inhibition of intestinal absorption of essential minerals such as $Ca^{2+}$, $Mg^{2+}$, and $Fe^{2+}$ in a concentration-dependent manner (*Figure 1G–H*, *Figure 1—figure supplement 1A*). Serum levels of $Ca^{2+}$ remained within the normal range except in rats fed 5% phytate, which were associated with slightly reduced serum $Ca^{2+}$ (*Figure 1I*). High-phytate-fed rats developed hypophosphatemia (*Figure 1J*) and hypomagnesemia (*Figure 1k*). Parathyroid hormone (PTH) levels markedly increased (5.4–11.7-fold) in the high-phytate groups by week 5 (*Figure 1*, *Figure 1—figure supplement 1B*) and remained elevated (2.8–6.4-fold) to week 12 (*Figure 1L*) in all phytate-fed rats because of sustained inhibition of intestinal $Ca^{2+}$ absorption. These results suggest that PTH plays a role in the development of hypophosphatemia in high-phytate-fed rats.

Serum levels of creatinine (Cr) remained within the normal range except in rats fed 5% phytate, in which Cr levels were increased compared with controls (*Figure 1*, *Figure 1—figure supplement 1C*). By contrast, serum levels of blood urea nitrogen (BUN) showed a marked increase in rats fed high levels of phytate (3% and 5%) compared with controls (*Figure 1M*). The BUN/Cr ratio is a common laboratory value used to diagnose azotemia and acute kidney injury (*Uchino et al., 2012*). Therefore, we determined the BUN/Cr ratio in phytate-fed rats. Rats exhibited a large increase in the BUN/Cr ratio in all phytate groups (*Figure 1M*, *Figure 1—figure supplement 1D*). Elevated BUN/Cr ratio may provide a biochemical signature of increased muscle catabolism associated with decreased fat-free lean mass in rats fed a high-phytate diet (*Haines et al., 2019*). These results indicate that a high-phytate diet may lead to the development of azotemia.

## Phytate-fed rats develop crystal nephropathies

To identify the cause of azotemia, we analyzed T2-weighted magnetic resonance images (MRI) of phytate-fed rats. MRI analysis revealed that rats fed 5% phytate had multiple renal cysts at the corticomedullary junction (*Figure 2A*, *Figure 2—figure supplement 1A*). The kidneys of these rats exhibited pronounced renomegaly with pitted, irregular, and granular cortices and medullar morphology (*Figure 2—figure supplement 1B–C*). The histopathological changes in the kidneys were evaluated while experimenters were blinded to the experimental group and scored for overall severity of renal injury, presence and severity of renal injury, $Ca^{2+}$ deposits, and fibrosis. The severity of renal injury was greatest in female rats fed 3% or 5% phytate and in male rats fed 5% phytate, whereas control rats lacked significant pathological changes.

The presence and severity of nephrocalcinosis matched the severity of renal injury scores (*Figure 2B–C*). Pathological changes in phytate-fed rats were consistent with mild to severe crystal nephropathies (*Mulay and Anders, 2017*; *Shavit et al., 2015*), including interstitial fibrosis, interstitial inflammation, tubular degeneration (ectasia, nephrocalcinosis, and protein casts), and enlarged or shrunken glomeruli (*Figure 2D–F*, *Figure 2—figure supplement 1C–F*). Alizarin Red S and von Kossa staining confirmed nephrocalcinosis, showing that the kidneys from phytate-fed rats contained extensive medullary $Ca^{2+}$ deposits (*Figure 2G*). Interstitial fibrosis in phytate-fed rats was further confirmed using Masson's trichrome staining and immunohistochemical analysis of α-smooth muscle

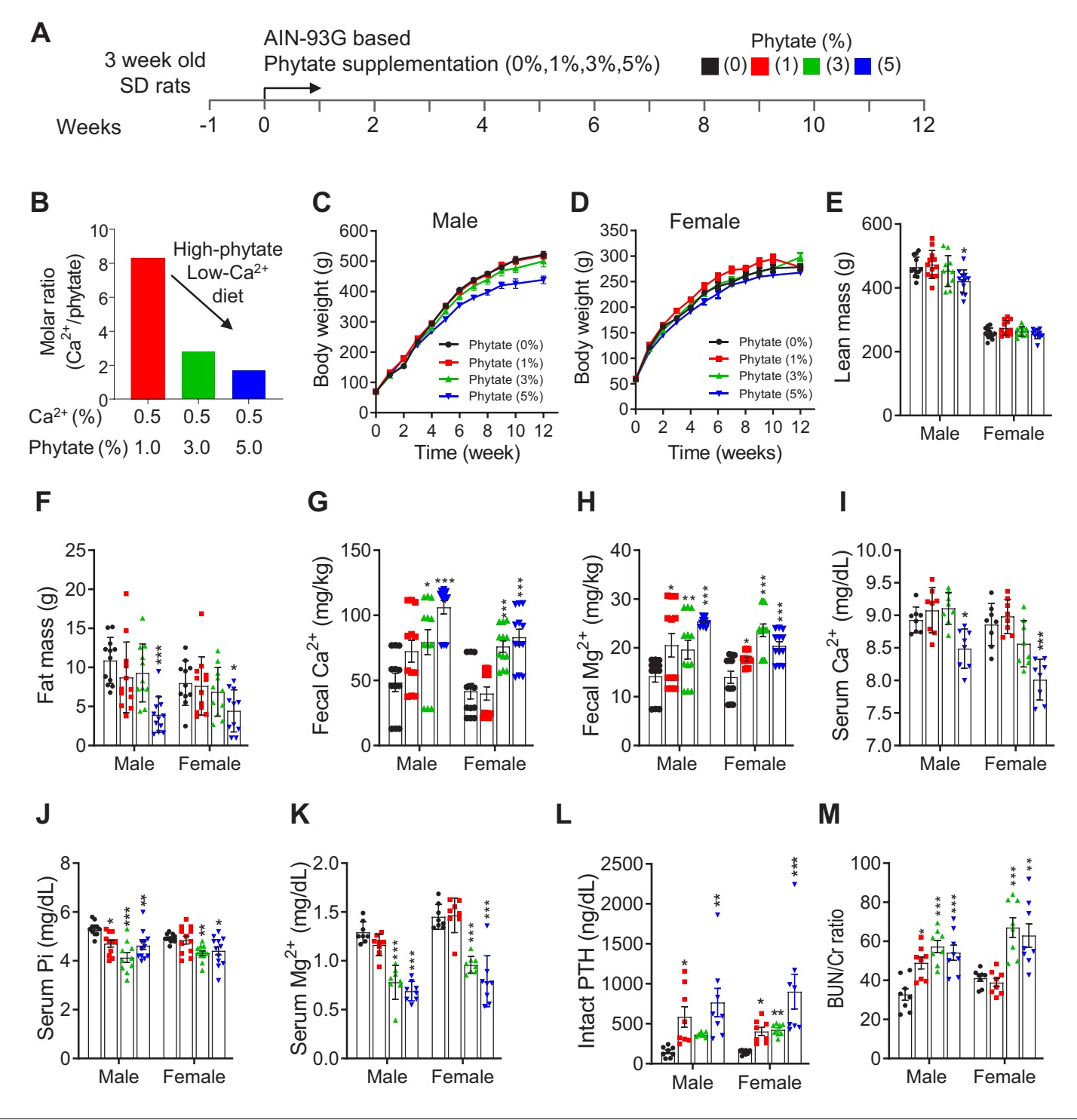

**Figure 1.** Phytate-fed rats develop secondary hyperparathyroidism with hypophosphatemia and azotemia. (**A**) Experimental design. After a 1-week period of adaptation with control AIN93G diet, we fed 4-week-old male and female Sprague–Dawley rats with AIN-93G diets supplemented with 0% (control), 1%, 3%, or 5% phytate for 12 weeks. (**B**) Molar ratio of $Ca^{2+}$ to phytate content in the AIN93G diet (0.5% $Ca^{2+}$) supplemented with 1%, 3%, and 5% phytate. (**C–D**) Mean body weights of male (**C**) and female (**D**) rats of all groups after 12 weeks of feeding with the experimental diets. (**E–F**) Mean whole-body lean mass (**E**) and fat mass (**F**) values for all groups. Whol-body lean mass and fat mass were measured by nuclear magnetic resonance spectroscopy. (**G–H**) ICP-MS analysis of fecal excretion of $Ca^{2+}$ (**G**) and $Mg^{2+}$ (**H**) in all groups after 3 weeks of feeding with the experimental diets. (**I–M**) Serum levels of $Ca^{2+}$ (**I**), phosphate (**J**), $Mg^{2+}$ (**K**), intact PTH (**L**), and blood urea nitrogen (BUN)/Cr ratio (**M**) of all groups after 12 weeks of feeding with the experimental diets. Serum intact PTH levels were measured by enzyme-linked immunosorbent assay (ELISA). All comparisons were

*Figure 1 continued on next page*

*Figure 1 continued*

performed by one-way ANOVA with Tukey's post hoc multiple comparison test or the Kruskal–Wallis test (G, H, L) for non-parametric statistical analysis of data that did not have a normal distribution. The groups in panels G and H and 5% phytate in female rats failed the normality test, but no outliers were identified. Panel L failed the normality test, one sample of 3% phytate in male rats was out of range according to ROUT (Q = 1%), but it did not alter the statistical outcome and no data were excluded from that group. All data are presented as the mean ± standard deviation (SD) for each group ($n$ = 8 – 12 per group). *$p$ < 0.05, **$p$ < 0.01, ***$P$ < 0.001 compared with controls.

The online version of this article includes the following figure supplement(s) for figure 1:

**Figure supplement 1.** Metabolic characteristics of Sprague-Dawley rats fed control and phytate-supplemented diets.

actin, a marker of fibrosis (*Figure 2H–I*). Together, the histopathologic data demonstrate that a high-phytate diet resulted in marked renal injury with overt nephrocalcinosis and crystal nephropathy.

## Phytate-fed rats develop severe bone loss

Growth retardation and metabolic bone disease are common in children and adolescents who have renal disease (*Wesseling-Perry et al., 2012*). To determine whether the phytate-fed rats developed skeletal changes and bone pathology coincident with renal disease, we performed dual-energy X-ray absorptiometry (pDEXA) and histological analyses of excised femora. Phytate-fed rats exhibited marked decreases in the rates of bone accrual along with decreased bone mineral density (BMD) in a concentration-dependent manner when compared with controls (*Figure 3A*). μ-CT images revealed concomitant increases in the bone marrow cavity space in phytate-fed rats compared with controls (*Figure 3B*). Further, phytate-fed rats exhibited significantly decreased trabecular bone volume (BV/TV), trabecular bone surface (BS/TV), trabecular number (Tb.N), and significantly increased trabecular space (Tb.Sp) compared with controls (*Figure 3C*). Consistent with the trabecular bone phenotypes, phytate-fed rats had markedly reduced cortical bone area (Tt.Ar) and increased cortical area fractions (Ct.Ar/Tt.Ar) (*Figure 3D*).

Male and female rats exhibited the same skeletal alterations, so we characterized the bone histopathology of the latter. Bone sections stained with toluidine blue showed that high-phytate diets decreased cartilage growth-plate thickness and bone osteoids (*Figure 3E*), consistent with skeletal alterations. Serum levels of receptor activator of nuclear factor kappa-B ligand (RANKL), a key component of osteoclastogenesis (*Kong et al., 1999*), were markedly increased in high-phytate-fed rats compared with controls (*Figure 3F*). The levels of osteoprotegerin (OPG), a RANKL decoy receptor that inhibits osteoclast differentiation and activation (*Lacey et al., 1998*), were significantly reduced (*Figure 3G*), and the sRANKL/OPG ratio was significantly increased in phytate-fed rats (*Figure 3H*). These data suggest a systemic acceleration of osteoclastogenesis in response to dietary phytate. Analysis of bone sections stained for tartrate-resistant acid phosphatase (TRAP) revealed that consumption of the high-phytate diet resulted in increased TRAP⁺ multinucleated cells (*Figure 3I*). Further, bone marrow-derived stromal cell cultures exhibited a significant concentration-dependent increase in the number of giant TRAP⁺ multinucleated osteoclasts (*Figure 3J*). Together, the data led us to posit that a high-phytate diet contributed to severe bone loss.

## High-phytate diets dysregulate mineral and phosphate metabolism in humans

To determine whether our findings using rats were applicable to humans, we conducted a non-comparative pilot study to evaluate the short-term effects of phytate on six healthy premenopausal women (aged 23–34 years, body mass index, 17.7–25.8 kg/m$^2$) (*Figure 4A*). The 12 day trial consisted of three consecutive diet cycles (four days in each cycle) of (1) polished white rice (WR, 0.35% phytate); (2) brown rice (BR, 1.07% phytate); and (3) brown rice with rice bran (BR+RB, 2.18% phytate; *Supplementary file 2*). At the end of each diet cycle, 16 hr fasting blood and 24 hr urine samples were collected from each participant. Serum levels of Ca$^{2+}$, magnesium, and phosphate remained within the normal range for all three diets, although high dietary phytate intake was associated with slightly decreased serum phosphate levels (*Figure 4B–C*). Results for other biochemical parameters are summarized in *Supplementary file 3*.

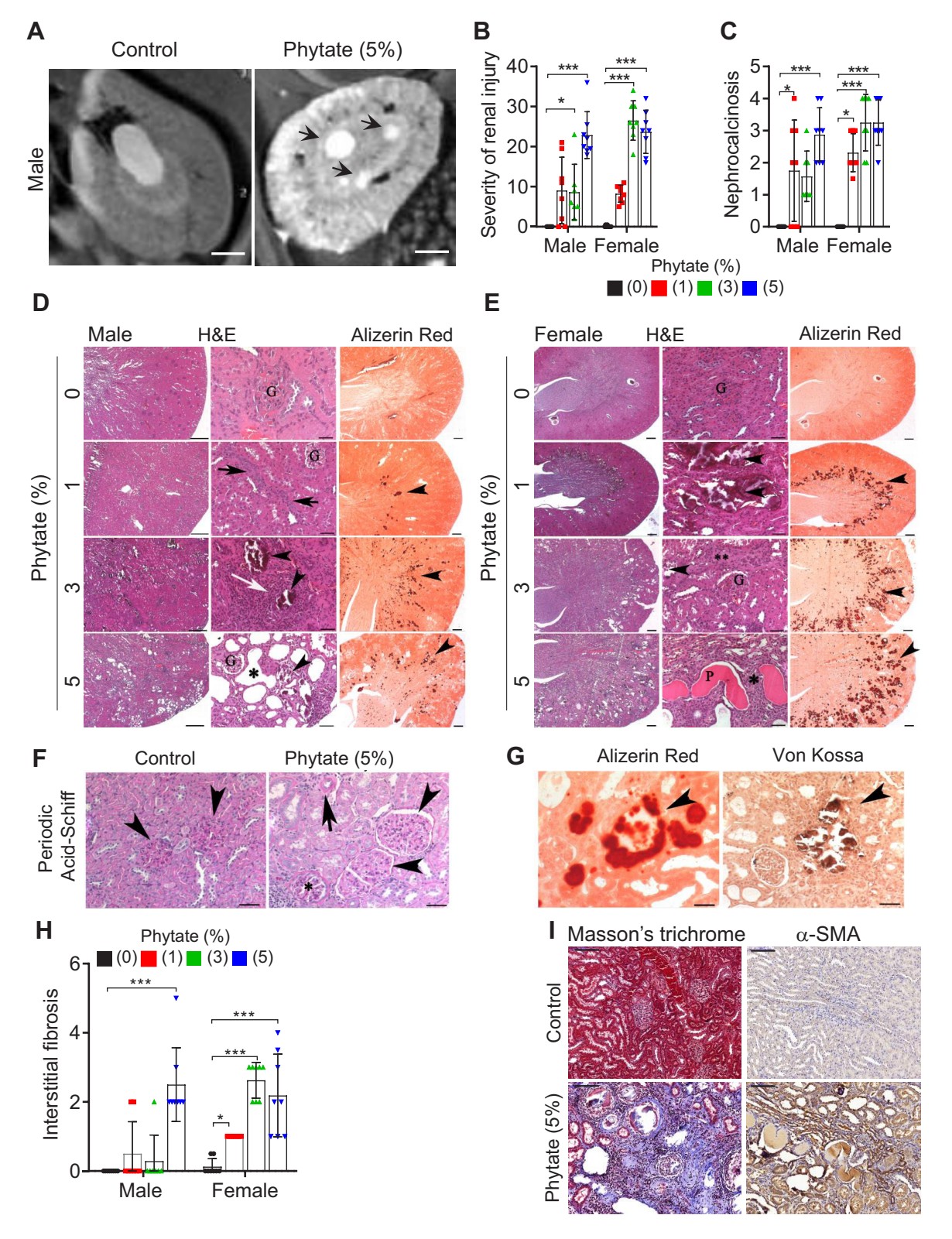

**Figure 2.** Phytate-fed rats develop crystal nephropathies. (**A**) Representative T2-weighted magnetic resonance images (MRIs) of kidneys from male rats fed a 5% phytate or control diet. The arrows point to kidney cysts in a phytate-fed rat. Scale bars, 250 μm. (**B–E**) Kidney injury was examined by staining with hematoxylin and eosin, Alizarin Red S (AR), von Kossa (VK), or periodic acid-Schiff (PAS) staining. Scale bars in panels (**D, E**): left, 500 μm; center, 50 μm; right, 500 μm. Granular material within the tubules (nephrocalcinosis) (arrowheads in panels [**D, E**]) was present in the kidneys of phytate-fed rats

*Figure 2 continued on next page*

**Figure 2 continued**

visualized using Alizarin Red S staining (D, E, right panels). Pathologic changes included tubular regeneration (black arrows), inflammation (white arrow), ectatic tubules (*), protein casts ('P'), interstitial fibrosis (**), and small shrunken glomeruli (G). (F) Representative PAS-stained sections from control (left) or phytate-fed (right) groups. (G) Calcification was detected using Alizarin Red S (left) and VK (right) stains. VK staining for $Ca^{2+}$ was similar to that of Alizarin Red S (arrowheads). In phytate-fed rats with severe pathologic changes, glomeruli were often large (5% phytate, arrowheads) with increased PAS-positive material in the mesangium, Bowman's capsule of the glomeruli (*), and in the basement membrane of the tubules (arrow). Kidneys from control rats were unremarkable. Scale bar, 50 μm. (H, I) Interstitial fibrosis was further detected by staining with Masson's trichrome (H) or by immunohistochemistry using antibodies against α-smooth muscle actin (I). Scale bar, 100 μm. All comparisons were performed by one-way ANOVA, with Tukey's post hoc multiple comparison test or the Kruskal–Wallis test (B, C, H) for non-parametric statistical analysis of data that did not have a normal distribution. The groups in panels (G) and (H), as well as the 5% phytate in female rats, failed the normality test, but no outliers were identified. Panel (L) failed the normality test and one sample of 3% phytate in male rats was out of range according to ROUT (Q = 1%), but it did not alter the statistical outcome and no data were excluded from that group. All data are presented as the mean ± standard deviation (SD) for each group (n = 8–12 per group). *, p<0.05; **, p<0.01; ***, p<0.001 compared with controls.

The online version of this article includes the following figure supplement(s) for figure 2:

**Figure supplement 1.** Histopathological features of the kidneys of phytate-fed rats.

Urinary pH levels decreased in a dose-dependent manner as the intake of phytate increased (*Figure 4D*). Higher dietary phytate intake resulted in a marked decrease in the 24 hr urinary $Ca^{2+}$ creatinine clearance rate (*Figure 4E*), whereas the 24 hr urinary phosphate creatinine clearance rate was only slightly increased (*Figure 4F*). These findings are consistent with the results of the tubular reabsorption rate, where high dietary phytate increases the renal tubular $Ca^{2+}$ reabsorption rate (*Figure 4G*) but decreases the renal tubular phosphate reabsorption rate (*Figure 4H*). These metabolic characteristics are probably secondary to the increased levels of PTH that occur in response to high levels of dietary phytate intake (*Figure 4I*). Further, high dietary phytate decreased the serum levels of FGF23 (*Figure 4—figure supplement 1A*). Thus, we propose that the high dietary phytate intake dysregulates mineral and phosphate metabolism in humans in a manner consistent with the findings of our rat studies.

## $Ca^{2+}$ supplementation protects phytate-fed rats from renal phosphate wasting through the formation of indigestible $Ca^{2+}$-phytate salts

Our study showed that diets containing >2–3% phytate dysregulate $Ca^{2+}$ and phosphate homeostasis in rats as well as in humans. Thus, when we calculated the daily amount of consumption of 3% phytate in rats, we found that administering 150–300 mg/kg phytate per day was harmful to rats. This daily dose range of phytate may correspond to 24.2–48.4 mg/kg body weight of humans (1,451–2,900 mg/60 kg) according to a simple practice guide for dose conversion between animals and humans (*Nair and Jacob, 2016*). This calculated daily dose range of dietary phytate represents an average intake that is greater than four-fold higher than those in the United States and the United Kingdom (631–746 mg) (*Ellis et al., 1987*).

Phytate binds $Ca^{2+}$ with high affinity at physiological pH (*Kim et al., 2010*; *Reddy and Sathe, 2002*). Consequently, the detrimental effects of phytate only occur when a diet poor in trace elements or $Ca^{2+}$ is consumed (*Raboy, 2007*; *Humer et al., 2015*). We asked therefore whether increasing the molar ratio of $Ca^{2+}$/phytate can modulate the bioavailability and digestibility of phytate in vivo. To address this question, we fed female rats a diet of 3% phytate supplemented with high (1%) $Ca^{2+}$ (high phytate with high $Ca^{2+}$ [HP-$HCa^{2+}$]) to increase the molar ratio of $Ca^{2+}$/phytate and $Ca^{2+}$ bioavailability compared with that provided by 3% phytate supplemented with low (0.5%) $Ca^{2+}$ (high phytate with low $Ca^{2+}$ [HP-$LCa^{2+}$]) (*Figure 5A– Supplementary file 1*). The growth rates of rats fed the HP-$HCa^{2+}$ diet were decreased and not associated with significant changes in serum levels of $Ca^{2+}$, despite slightly higher food and water intake compared with controls (*Figure 5—figure supplement 1A–D*). Moreover, rats fed the HP-$HCa^{2+}$ diet completely normalized their serum levels of phosphate and BUN, as well as the BUN/Cr ratio compared with rats fed the HP-$LCa^{2+}$ diet (*Figure 5B–C*, *Figure 5—figure supplement 1E*), indicating that supplementation with high $Ca^{2+}$ ameliorated hypophosphatemia and azotemia.

To identify the mechanism of phytate-induced hypophosphatemia, we collected 24 hr urine samples from phytate-fed rats. Rats fed the HP-$LCa^{2+}$ diet had markedly decreased urine pH (*Figure 5— figure supplement 1F*) without significant differences in urine volume (*Figure 5—figure*

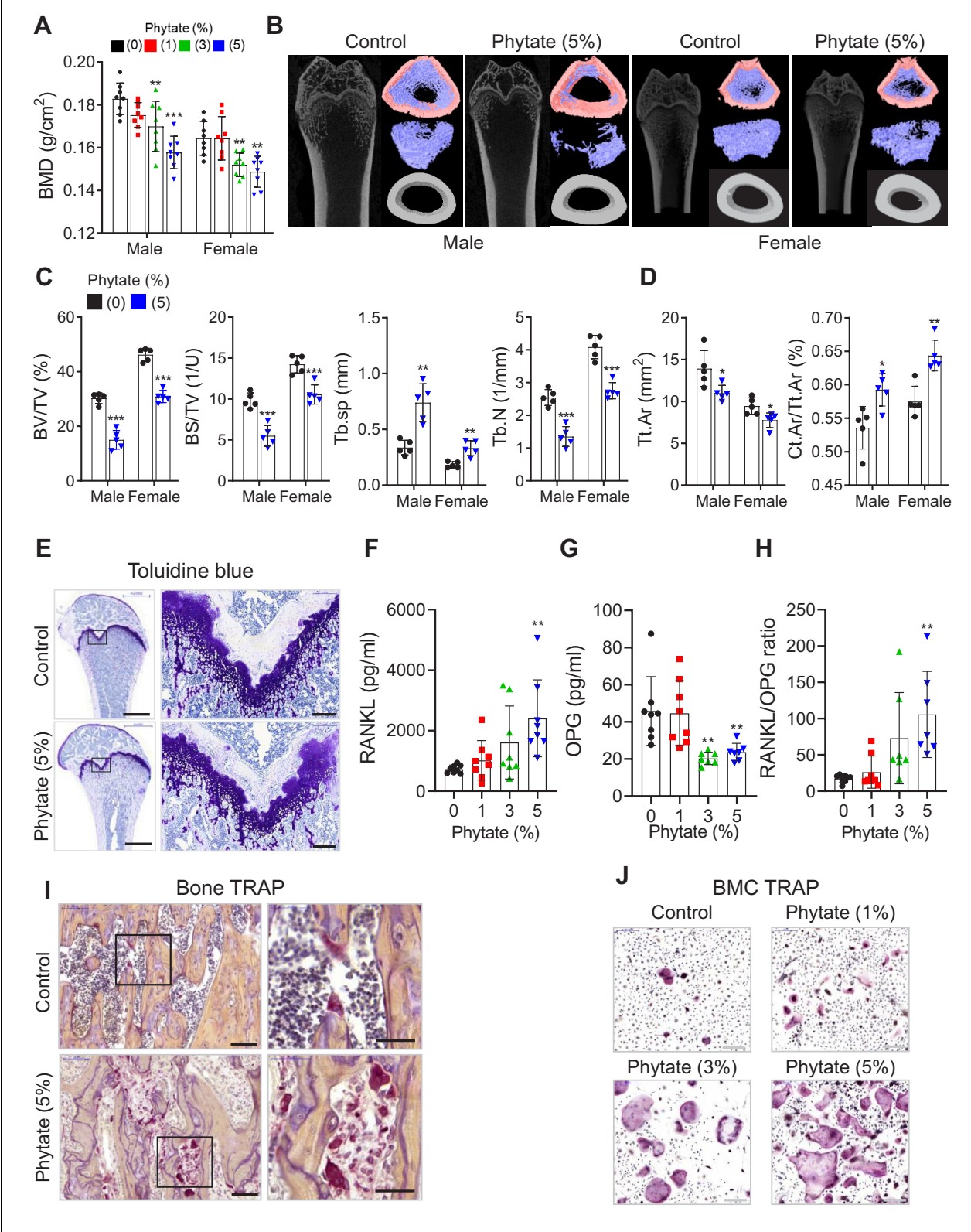

**Figure 3.** Phytate-fed rats develop severe bone loss. (**A**) Bone mineral density (BMD) was examined using peripheral dual-energy X-ray absorptiometry (pDEXA) of whole femora from rats fed control or phytate-supplemented diets (n = 8 per group). (**B**) Three-dimensional reconstructed μ-CT images of distal femora of rats fed control or phytate-supplemented diets. (**C**) Analysis of secondary spongiosa of the distal femur: bone volume fractions (BV/TV), bone surface density (BS/TV), trabecular separation (Tb.Sp), and trabecular number (Tb.N). (**D**) Analysis of the cortical bone of the distal femur: total

*Figure 3 continued on next page*

Figure 3 continued

cross-sectional area inside the periosteal envelope (Tt.Ar) and cortical area fraction (Ct.Ar/Tt.Ar). (E) Representative toluidine blue-stained sections of decalcified femora from female rats fed control or phytate-supplemented diets. Scale bars: left, 1000 µm; right, 100 µm. (F–H) Serum levels of soluble RANKL (F), OPG (G), and sRANKL/OPG ratio (H) in female rats fed control or phytate-supplemented diets. Serum sRANKL and OPG were measured using an ELISA (n = 8 per group). (I) Representative images of tartrate-resistant acid phosphatase (TRAP)-stained, EDTA-decalcified femur sections of rats fed control (upper) or phytate-supplemented (lower) diets. Red reaction product represents TRAP$^+$ multinucleated osteoclasts. Scale bars: left, 500 µm; right, 100 µm. (J) Bone marrow macrophages of rats fed control or phytate-supplemented diets for 12 weeks were cultured with RANK and macrophage colony-stimulating factor for 5 days. Osteoclasts were stained to detect TRAP activity. Scale bars: 200 µm. Comparisons were performed by unpaired Student's $t$ test (C, D), one-way ANOVA with Tukey's post hoc multiple comparison test (F–H), or the Kruskal–Wallis test (C) for non-parametric statistical analysis of data that did not have a normal distribution. The HP-LCa$^{2+}$ diet group in panel (C) failed the normality test, but no outliers were identified. All data are presented as the mean ± SD of each group ([C, D] 0% and 5% phytate, n = 5 per group; [F–H] n = 8 per group). *, p<0.05; **, p<0.01; ***, p<0.001 compared with controls.

supplement 1G), marked decreases in the 24 hr urinary Ca$^{2+}$/Cr ratio (Figure 5D), and a marked increase in the 24 hr urinary Pi/Cr ratio (Figure 5E) compared with controls. Further, rats fed the HP-LCa$^{2+}$ diet had markedly higher renal tubular reabsorption of Ca$^{2+}$ (TmCa$^{2+}$/GFR), with lower renal tubular reabsorption of phosphate (TmP/GFR) compared with controls (Figure 5F–G). Furthermore, rats fed the HP-LCa$^{2+}$ diet had a markedly increased 24 hr urinary protein/Cr ratio compared to controls, suggesting that a high-phytate diet contributes to the development of proteinuria, a sign of renal damage (Figure 5—figure supplement 1H). We surmised that the high-phytate diet decreased urinary Ca$^{2+}$ excretion while increasing renal phosphate wasting. Impaired renal Ca$^{2+}$ and phosphate reabsorption are restored in rats fed the HP-HCa$^{2+}$ diet (Figure 5F–G).

High-performance ion chromatographic (HPIC) analysis of feces demonstrated that rats fed the HP-HCa$^{2+}$ diet excreted an approximately three-fold higher level of fecal phytate compared with rats fed the HP-LCa$^{2+}$ diet (Figure 5H–I). ICP-MS analysis, performed after acid hydrolysis of fecal phytate to generate phosphate, further confirmed that rats fed the HP-HCa$^{2+}$ diet exhibited an approximately 3.8-fold greater increase in Ca$^{2+}$ and phosphate excretion compared with rats fed the HP-LCa$^{2+}$ diet (Figure 5J–K), confirming findings of increased fecal excretion of undigested Ca$^{2+}$-phytate in rats fed the HP-HCa$^{2+}$ diet. Thus, we conclude that rats fed the HP-HCa$^{2+}$ diet accumulated unabsorbed and indigestible Ca$^{2+}$-phytate salts in the intestine while decreasing the intestinal phosphate available for absorption, indicating that the endogenous enzyme phytase was unable to hydrolyze Ca$^{2+}$-phytate salts.

Conversely, rats fed the HP-LCa$^{2+}$ diet hydrolyzed and absorbed >68–74% of the phosphate in the intestine (Figure 5H–K) with a concomitant increase in urinary phosphate excretion, thus developing renal phosphate wasting (Figure 5E–G). These data indicate that rats that were fed the HP-LCa$^{2+}$ diet exhibited increased intestinal phosphate overloading caused by excessive phytate hydrolysis, which is consistent with evidence indicating that renal excretion of phosphate increases when phosphate intake is excessive (Vervloet et al., 2017). As phosphate and Ca$^{2+}$ uptake occur along the gastrointestinal tract via the dedicated phosphate channel NaPi-2b and the major Ca$^{2+}$ channel TRPV6, we analyzed the expression levels of NaPi-2b and TRPV6 in the intestinal tissues by immunohistochemical staining. However, there were no differences in expression levels of intestinal NaPi-2b or TRPV6 between any of the different dietary groups (Figure 5—figure supplement 2A–D). Thus, our findings demonstrate that phytate was digestible and absorbed in the intestine, considering that phytate is indigestible and unabsorbable in vivo when it forms Ca$^{2+}$-phytate salts in the presence of high Ca$^{2+}$ concentrations. Further, these findings are consistent with those of our previous in vitro studies (Kim et al., 2010).

## Changes in the microbiota of phytate-fed rats and the expression levels of intestinal multiple inositol polyphosphate phosphatase as a potential phytase

Given that phytate is digestible in the presence of low-Ca$^{2+}$ diets but is indigestible in the presence of high Ca$^{2+}$ diets in vivo, we examined whether phytate can be hydrolyzed by phytases from either intestinal microbiota or intestinal tissues. To address this question, we analyzed the predominant bacteria in rat feces by quantitative real-time PCR for 16S rDNA using a set of phylum- and class-specific primers (Supplementary file 4) for Firmicutes, Bacteroidetes, β-Proteobacteria, and

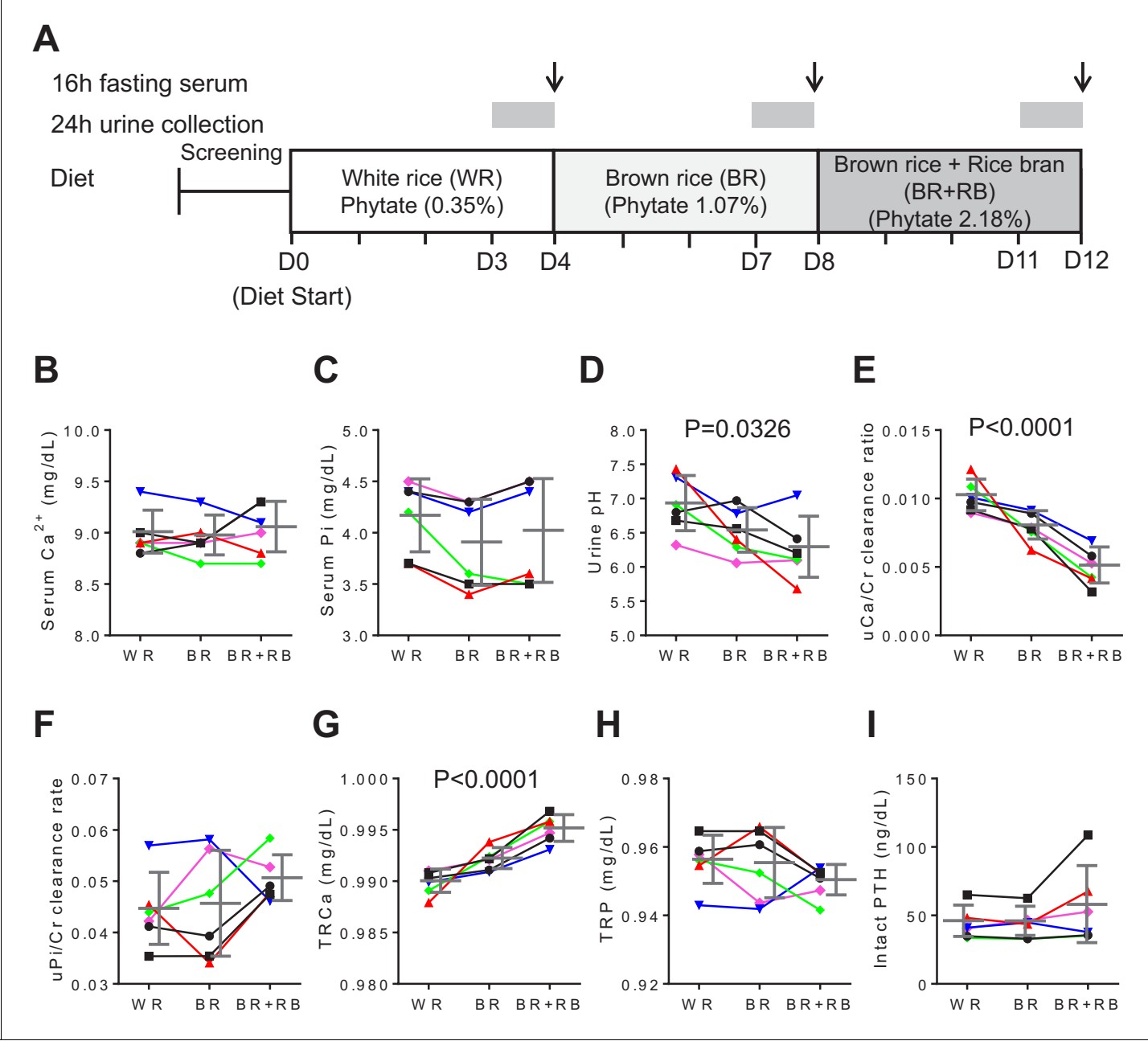

**Figure 4.** High-phytate diets dysregulate mineral and phosphate metabolism in humans. (A) Experimental design (six healthy female volunteers). (B, C) Time course of serum levels of $Ca^{2+}$ (B) and phosphate (C) for three different phytate doses. (D–H) Kinetics of urine pH (D), 24 hr urine $Ca^{2+}$/Cr ratio (E), 24 hr urine Pi/Cr ratio (F), renal tubular $Ca^{2+}$ reabsorption (TRCa) (G), and renal tubular phosphate reabsorption (TRP) (H) for three different phytate doses. (I) Time-course analysis of serum levels of intact PTH for different phytate doses. Data are presented as the mean ± SD for each dose (n = 6 per dose). Statistical significance was tested by repeated measure ANOVA between subjects and groups. All data are presented as the mean ± SD for each dose (n = 6 per dose).

The online version of this article includes the following figure supplement(s) for figure 4:

**Figure supplement 1.** High-phytate diets dysregulate mineral and phosphate metabolism in humans.

---

*Actinobacteria*. Phylum-level analysis indicated that rats fed the HP-LCa$^{2+}$ diet had significantly higher levels of *Firmicutes*, *Bacteroidetes*, and *β-Proteobacteria*, but had markedly lower levels of *Actinobacteria* compared to controls (***Figure 5—figure supplement 2E–H***). Interestingly, the relative abundance of *Bacteriodetes* and *β-Proteobacteria* had markedly increased, whereas the relative

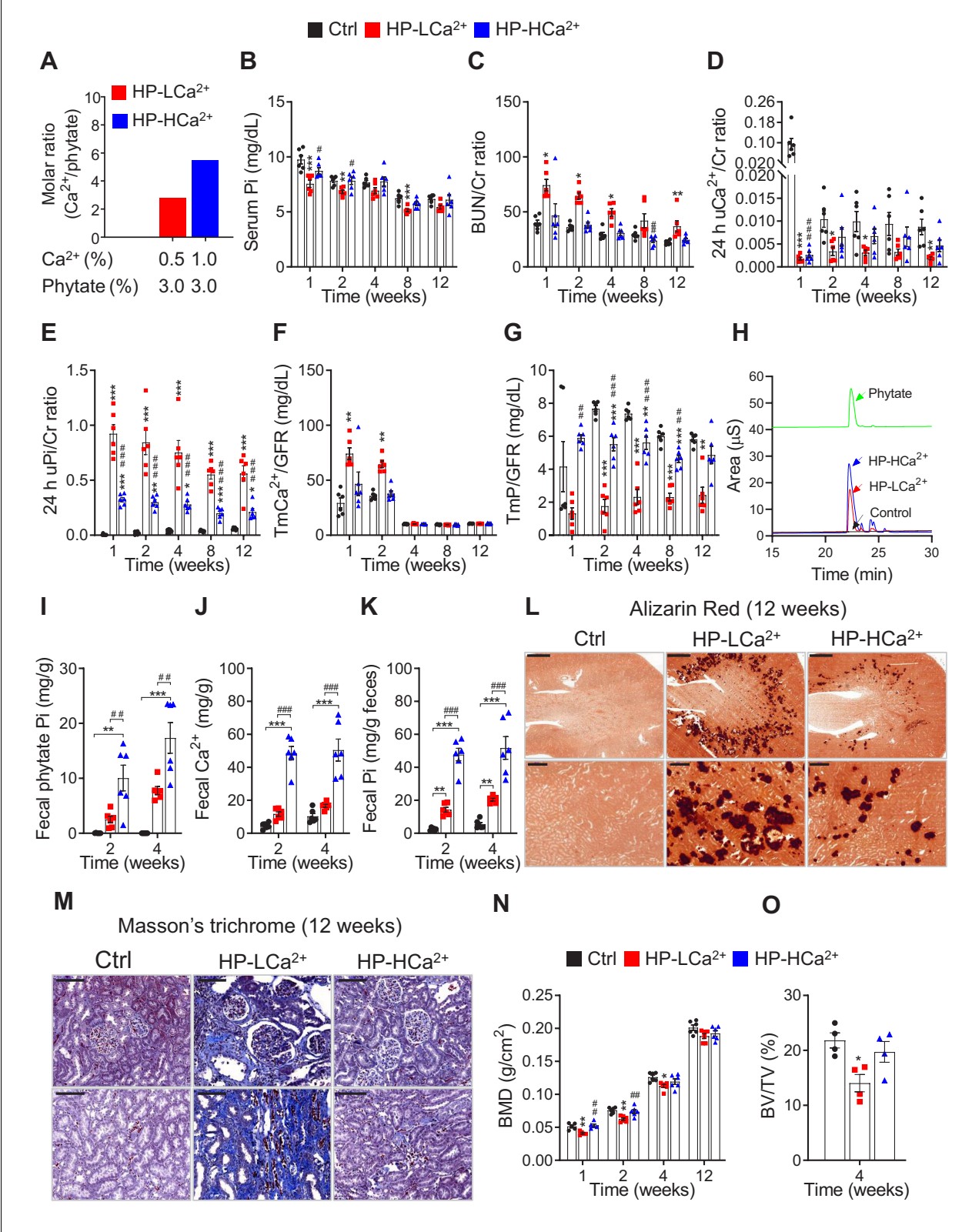

**Figure 5.** $Ca^{2+}$ supplementation protects phytate-fed rats from renal phosphate wasting disorders, crystal nephropathies, and bone loss via indigestible $Ca^{2+}$-phytate salts. (A) Molar ratio of $Ca^{2+}$ to phytate content when the diet was supplemented with 1.0% $Ca^{2+}$. (B–C) Time-course analysis of serum Pi level (B) and BUN/Cr ratio (C) in rats fed control, HP-LCa$^{2+}$, and HP-HCa$^{2+}$ diets for 12 weeks. (D–G) Time-course analysis of 24 hr urinary $Ca^{2+}$/Cr ratio (D), 24 hr urinary Pi/Cr ratio (E), ratio of tubular maximum reabsorption rate of renal tubular $Ca^{2+}$ to glomerular filtration rate (TmCa$^{2+}$/GFR) (F), and

*Figure 5 continued on next page*

*Figure 5 continued*

ratio of tubular maximum reabsorption of renal tubular phosphate to glomerular filtration rate (TmP/GFR) (G) in all groups during 12 weeks of experimental feeding. (H) Representative data of fecal phytate analysis by HPIC. (J–K) Time-course analysis of fecal $Ca^{2+}$ (J) and Pi (K) by ICP-MS in all groups after 2 and 4 weeks. Fecal phytate phosphorus was calculated as follows: [Phytate Pi] = [([Pi] from rats fed phytate diets) − ([Pi] from rats fed control diet)]. (L–M) Representative data of Alizarin Red S staining (L) and Masson's trichrome staining (M) of kidney sections in all groups after 12 weeks. (N–O) Bone mineral density (BMD) (N) and bone surface/volume ratio (BS/BV) (O) in all groups. All comparisons were conducted by one-way ANOVA with Tukey's post hoc multiple comparison test. All data are presented as the mean ± SD for each group (n = 4–6 per group). *, $p<0.05$; **, $p<0.01$; ***, $p<0.001$, compared with controls. #, $p<0.05$; ##, $p<0.01$; ###, $p<0.001$, compared with HP-$LCa^{2+}$ diet group.

The online version of this article includes the following figure supplement(s) for figure 5:

**Figure supplement 1.** $Ca^{2+}$ supplementation protects rats fed phytate from crystal nephropathies.

**Figure supplement 2.** The expression levels of intestinal MINPP1, TRPV6, and NaPi-2b and real-time PCR analysis of stool microbiota in rats fed control, HP-$LCa^{2+}$, and HP-$HCa^{2+}$ diets for 12 weeks.

**Figure supplement 3.** $Ca^{2+}$ supplementation protects rats fed phytate from crystal nephropathies.

**Figure supplement 4.** $Ca^{2+}$ supplementation protects phytate-fed rats from renal fibrosis and skeletal abnormalities.

abundance of *Firmicutes* had decreased in response to high-phytate diets (*Figure 5—figure supplement 2I*), suggesting that diets containing high-phytate altered the composition of the gut microbiota.

As we observed phytate hydrolysis in rats fed the HP-$LCa^{2+}$ diet, we further investigated whether phytate hydrolysis is associated with either microbial or mammalian phytase activity. Although humans and other monogastric animals, such as pigs, chickens, mice, and rats, are known to lack or have low phytase activities capable of hydrolyzing dietary phytate (*Iqbal et al., 1994*), recent studies and data in biological databases, such as UniProtKB and the Human Protein Atlas, have suggested that multiple inositol polyphosphate phosphatase 1 (MINPP1) is a strong candidate as a histidine acid phytase in humans and other monogastric animals as well as in gut microbiota (*Cho et al., 2006*; *Stentz et al., 2014*). MINPP1 could be a secreted enzyme with extremely high catalytic activity for phytate and is widely expressed in several tissues, including the stomach, duodenum, small intestine, colon, and pancreas, as well as at the highest levels in kidney, liver, and placenta. Phylogenic analysis of bacterial MINPP1 (*Stentz et al., 2014*) suggested that Bacteroidetes express and secrete a homolog of a mammalian MINPP1 in the gut, whereas Firmicutes, a major phylum of the human intestinal microbiota, do not encode a protein homolog of MINPP1 (*Supplementary file 5*). These results suggested that increased phytate hydrolysis in rats fed the HP-$LCa^{2+}$ diet is associated with the increase in relative abundance of *Bacteroidetes* and the decrease in relative abundance of *Firmicutes* (*Figure 5—figure supplement 2I*). Consistent with microbiota changes, immunohistochemical analysis of mammalian intestinal MINPP1 demonstrated that rats show high levels of MINPP1 expression in the villi of the intestine. Importantly, rats fed the HP-$HCa^{2+}$ diet had markedly increased levels of intestinal MINPP1 (*Figure 5—figure supplement 2J–K*), probably to increase the enzymatic activity against chronic indigestible $Ca^{2+}$-phytate in the presence of a high $Ca^{2+}$ concentration for 12 weeks. Taken together, these findings suggest crucial roles of gut bacterial and intestinal MINPP1 in the hydrolysis of dietary phytate, although further research is needed to determine whether the hydrolysis of phytate by MINPP1 is direct and/or indirect.

## $Ca^{2+}$ supplementation protects phytate-fed rats against crystal nephropathies and bone loss

We next examined the morphological and histopathological changes in bones and kidneys when rats were fed the HP-$LCa^{2+}$ diet. $Ca^{2+}$ supplementation protected these rats against the development of the expected severe gross and microscopic changes of renomegaly (*Figure 5—figure supplement 3A–B*) and medullary nephrocalcinosis (*Figure 5L*, *Figure 5—figure supplement 3D–F*), respectively, as revealed by Alizarin Red S staining. However, we found no differences in serum levels of oxalate between diet groups (*Figure 5—figure supplement 3G*), suggesting that phytate-induced nephrocalcinosis was mainly with medullary calcium deposits, consistent with the findings of renal histology, such as Alizarin Red S and von Kossa staining. Further, rats that were fed the HP-$HCa^{2+}$ diet expressed lower levels of four key markers (*Kronenberg, 2009*) of renal damage and fibrosis

(*NGAL*, *COL1A1*, *COL6A2*, and *MPG*) (*Figure 5—figure supplement 4A*) and exhibited decreased renal fibrosis revealed by Masson's trichrome staining to detect collagen (*Figure 5M*, *Figure 5—figure supplement 4B–C*). These data suggest that the HP-LCa$^{2+}$ diet was a key contributor to renal damage and fibrosis. Further, pDEXA and μ-CT analyses of excised femora confirmed that rats fed the HP-HCa$^{2+}$ diet were protected against the expected bone loss and demineralization observed in rats fed the HP-LCa$^{2+}$ diet (*Figure 5N–O*, *Figure 5—figure supplement 4D–G*). Together, these findings support our hypothesis that high dietary phytate intake leads to damage of renal tissues as well as impaired bone mass accrual, whereas Ca$^{2+}$ supplementation of the high-phytate diet has protective effects.

## Phytate-mediated Ca$^{2+}$ deficiency promotes vitamin D insufficiency and renal phosphate wasting independent of FGF23 expression

To identify the physiological mechanisms of high-phytate diets on vitamin D metabolism, we performed a time-course analysis of the serum levels of PTH, 25(OH)D, and activated vitamin D (1,25 [OH]$_2$D). First, we found that the serum levels of PTH were consistently elevated in rats fed the HP-LCa$^{2+}$ diet compared with control rats and rats fed the HP-HCa$^{2+}$ diet (*Figure 6A*). Second, we found that rats fed the HP-LCa$^{2+}$ diet in conjunction with high PTH had decreased serum levels of 25 (OH)D compared with controls (*Figure 6B*), leading to increased serum levels of 1,25(OH)$_2$D (*Figure 6C*). In rats fed the HP-HCa$^{2+}$ diet, the serum levels of 25(OH)D and 1,25(OH)$_2$D normalized, indicating that Ca$^{2+}$ supplementation inhibited the metabolism of serum 25(OH)D to 1,25(OH)$_2$D (*Figure 6B–C*). Consistent with these findings, Ca$^{2+}$ supplementation normalized the renal expression levels of *CYP27B1*, *CYP24A1*, and vitamin D receptor (*VDR*) (*Figure 6D–F*). Further, our findings, which are consistent with those of a previous study (*Clements et al., 1987*), demonstrate that phytate-mediated inhibition of intestinal Ca$^{2+}$ absorption as well as increased phosphate overload promoted the metabolization of 25(OH)D to 1,25(OH)$_2$D, suggesting that high-phytate diets deplete serum 25(OH)D, which mitigates the harmful effects of phytate.

To identify proteins that contributed to phytate-mediated impairment of renal tubular Ca$^{2+}$ and phosphate reabsorption, we analyzed the mRNA levels of key renal genes associated with Ca$^{2+}$ and phosphate homeostasis, including those associated with Ca$^{2+}$ channels, phosphate channels, and regulatory proteins (*Mulay and Anders, 2017*; *Huang et al., 2013*). We found that the renal expression of key genes associated with renal phosphate wasting, including those encoding Ca$^{2+}$-sensing receptor (CaSR), inositol 1,4,5-trisphosphate receptor 1 (ITPR1), αKlotho, sodium/phosphate cotransporters (NaPi-2a and NaPi-2c), sodium-hydrogen antiporter 3 regulator 1 (NHERF1), and phosphate-regulating neutral endopeptidase X-linked, was markedly decreased in phytate-fed rats (*Figure 6—figure supplement 1A*). Each of these genes is associated with the development of nephrocalcinosis and renal phosphate wasting (*Shavit et al., 2015*; *Huang et al., 2013*; *Karim et al., 2008*; *Monico and Milliner, 2012*).

Immunoblot analysis further confirmed that the renal expression of αKlotho was significantly decreased in rats fed the HP-LCa$^{2+}$ diet compared with controls (*Figure 6G–H Figure 6—figure supplement 1B–C*), suggesting that low levels of renal αKlotho may represent a complementary mechanism that increases serum phosphate levels in response to severe hypophosphatemia in rats fed the HP-LCa$^{2+}$ diet. High serum FGF23 levels in conjunction with elevated levels of PTH contribute to the pathogenesis of renal phosphate wasting and hypophosphatemic rickets (*Vervloet et al., 2017*; *Rodriguez-Ortiz et al., 2012*; *Walton and Bijvoet, 1975*). Therefore, we analyzed the serum levels of FGF23. Unexpectedly, quantitative analysis did not detect significant changes in the serum levels of full-length FGF23 or its C-terminal FGF23 in rats fed the HP-LCa$^{2+}$ diet compared with controls (*Figure 6—figure supplement 1D–E*). Immunoblot and immunohistochemical analyses of sections of the femur revealed that rats fed the HP-LCa$^{2+}$ diet showed decreased levels of FGF23 in the calvarial bones (*Figure 6I–J*, *Figure 6—figure supplement 1F*), suggesting that the phytate-mediated renal phosphate wasting was mainly caused by high PTH levels independent of FGF23.

## Discussion

Here, we show that the HP-LCa$^{2+}$ diet was a significant risk factor for phosphate overloading and renal phosphate wasting associated with increased risks of nephrocalcinosis, renal fibrosis,

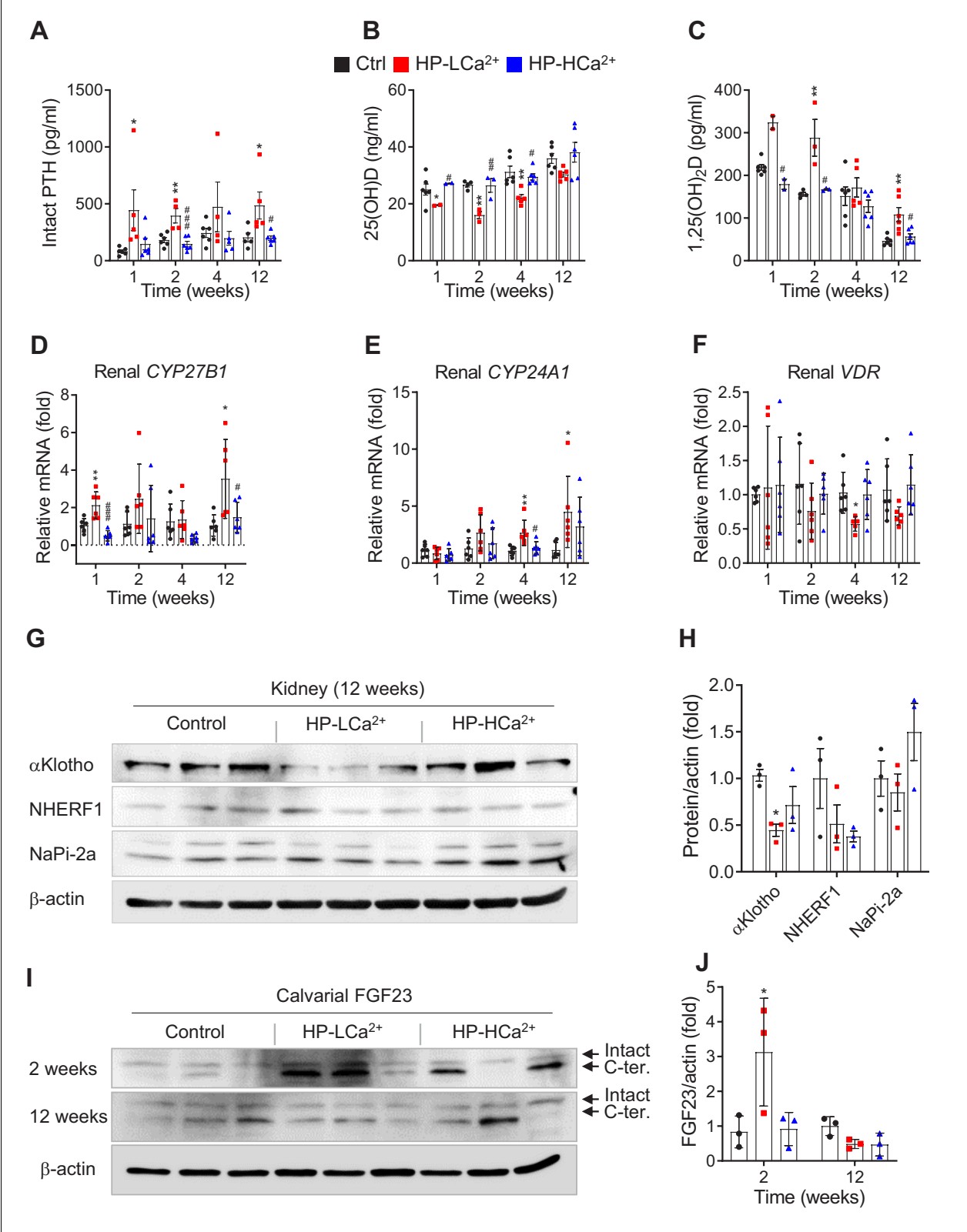

**Figure 6.** Phytate-mediated $Ca^{2+}$ deficiency promotes vitamin D insufficiency and renal phosphate wasting independent of FGF23 expression. (A–C) Time-course analysis of serum levels of intact PTH (A), 25(OH)D (B), and 1,25(OH)$_2$D (C) in rats fed control, HP-LCa$^{2+}$, and HP-HCa$^{2+}$ diets. For the early measurement of 25(OH)D (B) and 1,25(OH)$_2$D, we pooled the sera from 2 to 3 rats. (D–F) Time-course analysis of renal CYP27B1 (D), CYP24A1 (E), and VDR (F) in rats fed control, HP-LCa$^{2+}$, and HP-HCa$^{2+}$ diets. (G) Immunoblot analysis of renal αKlotho, NHERF1, NaPi-2a in rats fed control, HP-LCa$^{2+}$,

*Figure 6 continued on next page*

*Figure 6 continued*

and HP-HCa$^{2+}$ diets. (H) The levels of renal proteins were quantified using ImageJ software (NIH, Bethesda, MD, USA). (I) Immunoblot analysis of calvarial FGF23 in rats fed control, HP-LCa$^{2+}$, and HP-HCa$^{2+}$ diets. (J) The levels of total calvarial FGF23 were quantified using ImageJ software. All comparisons were conducted using two-way ANOVA with Tukey's post hoc multiple comparison testing or the Kruskal–Wallis test (A–C) for non-parametric statistical analysis of data that did not have a normal distribution. Panels (A–C) failed the normality test, but no outliers were identified. All data are presented as the mean ± SD of each group (n = 4–6 per group). *, p<0.05; **, p<0.01; ***, p<0.001, compared with controls.

The online version of this article includes the following figure supplement(s) for figure 6:

**Figure supplement 1.** High phytate-induced renal phosphate wasting is independent of FGF23.

crystal nephropathy, hypophosphatemia, and bone loss. Our findings support our hypothesis that an HP-LCa$^{2+}$ diet has detrimental metabolic effects on Ca$^{2+}$ and phosphate homeostasis in rats through inhibiting intestinal Ca$^{2+}$ absorption while increasing intestinal phosphate overloading. These events lead to high levels of PTH and low levels of FGF23 and αKlotho. The resulting hypocalcemia and hypophosphatemia caused by the high-phytate/low-Ca$^{2+}$ diet could explain

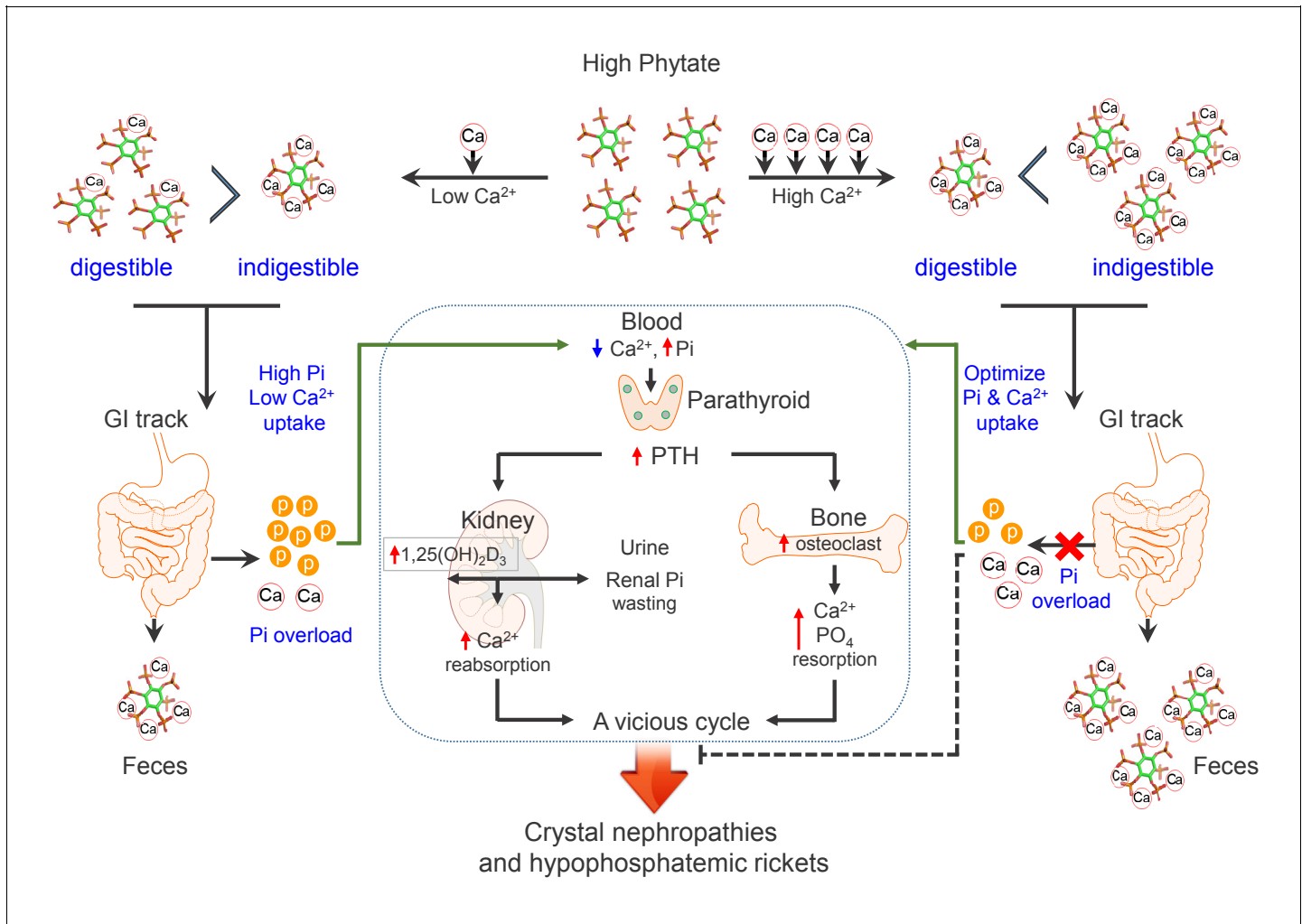

**Figure 7.** Model for the effects of phytate diets supplemented with low-Ca$^{2+}$ or high-Ca$^{2+}$ on Ca$^{2+}$ and phosphate homeostasis. In the presence of a low-Ca$^{2+}$ diet (Ca$^{2+}$/phytate <5), phytate can adversely affect Ca$^{2+}$ and phosphate homeostasis in normal healthy rats by inhibiting intestinal Ca$^{2+}$ absorption while increasing phosphate overload through phytate hydrolysis in the intestine. These effects lead to crystal nephropathies, severe renal phosphate wasting, and pathologic bone loss via increased PTH and 1,25(OH)$_2$D$_3$. By contrast, in the presence of a high-Ca$^{2+}$ diet, phytate forms multiple Ca$^{2+}$-phytate salts, which makes them indigestible and unabsorbable in vivo, leading to their excretion in the feces as undigested Ca$^{2+}$-phytate salts. This process prevents a vicious cycle of intestinal phosphate overloading and renal phosphate wasting while improving intestinal Ca$^{2+}$ bioavailability, which eventually alleviates the detrimental effects of excess phytate hydrolysis.

the secondary hyperparathyroidism, bone loss and phosphaturia, which in turn causes nephrocalcinosis. By contrast, supplementation with high $Ca^{2+}$ alleviated phytate-mediated pathogenic defects such as biochemical abnormalities, high PTH levels, nephrocalcinosis, renal fibrosis, vitamin D insufficiency, hypophosphatemia, and bone loss. We identified the dietary conditions that influence the bioavailability of phytate when an excess of $Ca^{2+}$ is added to high-phytate diets. Thus, phytate forms insoluble $Ca^{2+}$-phytate salts in the intestine, where it becomes indigestible and unabsorbable, and ultimately is excreted as undigested $Ca^{2+}$-phytate salts in the feces. Thus, $Ca^{2+}$ supplementation normalized impaired renal tubular $Ca^{2+}$ and phosphate reabsorption by preventing intestinal phosphate overloading caused by phytate hydrolysis and improving intestinal $Ca^{2+}$ bioavailability (*Figure 7*).

We identified previously unrecognized mechanisms of bioavailability and digestibility of phytate in vivo. Thus, HPIC and ICP-MS analyses of feces revealed that phytate is present in digestible or indigestible forms in vivo depending on the dietary $Ca^{2+}$ content, revealing that phytate is only indigestible and unabsorbable when it forms multiple $Ca^{2+}$-phytate salts. This finding is consistent with data of our in vitro studies (*Kim et al., 2010*), which show that phytate possesses three or four $Ca^{2+}$-binding sites at physiological pH and forms insoluble multiple $Ca^{2+}$-phytate salts. By contrast, when the $Ca^{2+}$/phytate molar ratio is <5 (HP-LCa$^{2+}$ diet), phytate becomes digestible, which leads to phosphate overloading in the intestine leading to renal phosphate wasting and metabolic bone disease. These results may explain why the adverse effects of high-phytate on nutrients are exacerbated when $Ca^{2+}$ intake or mineral content is low (*Raboy, 2007*; *Humer et al., 2015*).

Phytases can be classified into two major classes on the basis of their biochemical and structural properties, that is, the histidine acid phytases and the alkaline β-propeller phytases (*Oh et al., 2004*; *Stentz et al., 2014*). The histidine acid phytase (HAP) superfamily consists of purple acid phosphatase, histidine acid phytase, histidine acid phosphatase, and MINPP1 (*Supplementary file 5*). Our in vivo data suggested that a high-phytate diet can be hydrolyzed and can release excess phosphates into the gut by either mammalian intestinal or microbial MINPP1 when the diet contains a low concentration of $Ca^{2+}$, although further studies are needed to determine how mammalian MINPP1 is processed and secreted into the digestive tract. The HAPs, predominantly, have positively charged amino acids in their active sites, thereby explaining how the HAPs can bind highly negatively charged phytate at acidic pH but that this is unfavorable for binding to positively charged $Ca^{2+}$-phytate due to electrostatic repulsion between the $Ca^{2+}$-phytate and the active sites (*Oh et al., 2004*). This suggests a mechanism to explain why the HAPs can only hydrolyze $Ca^{2+}$-free phytate but cannot hydrolyze $Ca^{2+}$-phytate when the diet contains a high concentration of $Ca^{2+}$, leading to a decrease in the bioaccessibility of $Ca^{2+}$-phytate and the bioavailability of intestinal phosphates and $Ca^{2+}$ for absorption into the blood, eventually leading to excretion of undigested $Ca^{2+}$-phytate in the feces. These results explain why high-phytate/high-$Ca^{2+}$ diets prevented the development of crystal nephropathies, renal phosphate wasting, and bone loss in rats. By contrast, β-propeller phytases mainly have negatively charged amino acids in their active sites (*Ha et al., 2000*), which provide a favorable electrostatic environment and strict substrate specificity for the hydrolysis of positively charged $Ca^{2+}$-phytate but not $Ca^{2+}$-free phytate, due to electrostatic repulsion between negatively charged phytate and negatively charged amino acids of the active sites (*Supplementary file 5*). Therefore, we propose that humans and other monogastric animals may lack β-propeller phytases that hydrolyze $Ca^{2+}$-phytate in the gut, as high-phytate/high-$Ca^{2+}$ diets could not be hydrolyzed (indigestible) in the guts of the rats and, eventually, most $Ca^{2+}$-phytate was excreted in the feces.

Further, the potential toxicity of excess phosphate in the context of mineral bone disorders (MBD) associated with chronic kidney disease (CKD) (*Vervloet et al., 2017*; *Ketteler et al., 2017*) and $Ca^{2+}$-phosphate crystals (*Kuro-o, 2013*), together with our present findings, indicate that a HP-LCa$^{2+}$ diet can be particularly harmful to patients suffering from these conditions. Moreover, dietary phosphate overload can cause tubular damage, hypophosphatemia, and $Ca^{2+}$-phosphate crystals; and phosphate overloading never causes persistent hyperphosphatemia unless renal function is critically impaired (*Haut et al., 1980*). Thus, our study emphasizes that a detailed understanding of the dietary effects and properties of phytate may provide a new opportunity to reduce intestinal phosphate overload and to benefit patients with CKD, mineral bone disease, or crystal nephropathies.

Several lines of evidence support our findings that phytate is digestible in the absence of $Ca^{2+}$ or divalent minerals. First, studies of rats fed $^{14}C$-labeled phytate show that approximately 94% of phytate is absorbed when $Ca^{2+}$ intake is low, whereas 54% of phytate is excreted in the feces when $Ca^{2+}$ intake is high (*Nahapetian and Young, 1980*), suggesting that phytate absorption is modulated or inhibited by the $Ca^{2+}$ concentration. Second, another study of rats fed $^{3}H$-inositol phytate shows that 78% of phytate is absorbed and 14% is excreted in the feces. Further, most of the radioactivity represents *myo*-inositol or *myo*-inositol monophosphate in the plasma or urine (*Sakamoto et al., 1993*), suggesting the phytate is digestible in vivo. The biased belief that phytate is indigestible led the authors of the above two studies to conclude that intact phytate is absorbed through the upper intestine, hydrolyzed within mucosal cells before entry into the blood, and then released as *myo*-inositol and phosphate into the blood or urine.

Subsequently, investigators who assumed that phytate is absorbed and can be detected in the blood or urine (*Grases et al., 2000*) were unable to detect phytate in serum or urine despite using a highly specific and sensitive mass spectrometry (*Letcher et al., 2008*). These studies support the conclusion that phosphate or *myo*-inositol generated by the hydrolysis of phytate is absorbed rather than phytate, which is unabsorbable. Numerous explanations account for the impermeability of intestinal membranes to phytate (*Reddy and Sathe, 2002*). Phytate possesses six phosphate groups with strong negative charges at physiological pH, which suggests that phytate cannot cross the lipid bilayer of the plasma membrane without a specific receptor (*Ozaki et al., 2000*). In support of this possibility, we found that phytate is only engulfed by hepatic Kupffer cells and tissue macrophages in vivo when we intravenously injected iron-containing phytate particles into live mice (*Kim et al., 2017*; *Lee et al., 2018*), indicating that phytate cannot penetrate membranes of epithelial cells in the intestine. Together, these findings suggest that phytate is not absorbed in the intestine and that only products generated through the hydrolysis of phytate, phosphate, or *myo*-inositol can be absorbed in the intestine of rats fed phytate-containing low-$Ca^{2+}$ diets.

Here, we demonstrate that phytate can act as a dual demon in healthy normal rats by adversely affecting $Ca^{2+}$ and phosphate homeostasis through inhibition of intestinal $Ca^{2+}$ absorption while increasing intestinal phosphate overload, depending on the $Ca^{2+}$ content of diets. This may explain why rats fed the HP-LCa$^{2+}$ diet exhibited high levels of PTH and 1,25(OH)$_2$D, as well as low levels of FGF23 and αKlotho. Moreover, our findings are consistent with data from a large population-based study of subjects with preserved renal function without CKD, which showed that PTH levels increase earlier than FGF23 levels as kidney function decreases and that high FGF23 levels are more closely associated with low levels of 1,25(OH)$_2$D than with excessive renal phosphate wasting (*Dhayat et al., 2016*). Thus, our results suggest that high levels of 1,25(OH)$_2$D may suppress the expression of calvarial FGF23 in rats fed the HP-LCa$^{2+}$ diet, in accordance with an earlier study showing that $Ca^{2+}$ deficiency is highly associated with low levels of FGF23 (*Rodriguez-Ortiz et al., 2012*). Moreover, $Ca^{2+}$ deficiency increases the levels of 1,25(OH)$_2$D, which leads to low levels of FGF23 in $Ca^{2+}$-deficiency states (*Clements et al., 1987*).

However, previous studies that investigated the effects of phytate on kidney stones in rats report conflicting results, which we ascribe to differences in dietary compositions, particularly to the ratio of $Ca^{2+}$/phytate and analytical laboratory methods (*Grases et al., 2000*). Specifically, the AIN-76A purified rodent diet containing 1% phytate prevents the development of kidney stones compared with the AIN-76A controls (*Grases et al., 2000*). Further, *Grases et al. (2000)* concluded that rats fed the AIN-76A diet developed mineral deposits at the corticomedullary junction from Alizarin Red S staining, whereas *Lien et al. (2001)* did not find an increased incidence or severity of renal tubular mineralization in rats fed the AIN-76A or AIN-93G diet. Consistent with this, we found that rats fed the AIN-93G control diet did not have nephrocalcinosis. Moreover, we found that an AIN-93G-based HP-LCa$^{2+}$ diet induced biochemical and urinary abnormalities associated with kidney disease as well as histopathological defects in the kidney. Phytate in the AIN-96-G diet caused a dramatic time- and concentration-dependent increase in PTH levels, leading to PTH-induced impairment of the renal reabsorption of $Ca^{2+}$ and phosphate, and predisposed rats to the development of hypophosphatemic rickets and nephrocalcinosis.

In conclusion, we demonstrate here that a high-phytate low-$Ca^{2+}$ diet is a risk factor for nephrocalcinosis, renal inflammation, and fibrosis as well as renal phosphate wasting disorders and bone loss in rats. Our studies provide an animal model for studying the pathogenesis of crystal

nephropathies and bone loss associated with renal phosphate wasting, and further suggest that the restriction of phytate intake may reduce the risks of developing crystal nephropathies.

# Materials and methods

## Key resources table

| Reagent type (species) or resource | Designation | Source or reference | Identifiers | Additional information |
|---|---|---|---|---|
| Genetic reagent (*Rattus norvegicus*) | Sprague Dawley rat | Orient Bio, Seoul, Korea | | Developed by Sprague Dawley, Inc. |
| Chemical compound, drug | AIN-93G | Dyets Inc. | DYET# 10700 | Control diet |
| Chemical compound, drug | Phytic acid | Sigma-Aldrich | Cat. #: P-8810 | Phytate diet |
| Chemical compound, drug | HNO$_3$ | Sigma-Aldrich | Cat. #: 438073 | Fecal mineral analysis |
| Chemical compound, drug | HF | Sigma-Aldrich | Cat. #: 695068 | Fecal mineral analysis |
| Chemical compound, drug | HCl | Sigma-Aldrich | Cat. #: H1758 | Fecal mineral analysis |
| Chemical compound, drug | EDTA | Sigma-Aldrich | Cat. #: 93283 | Fecal phytate analysis |
| Chemical compound, drug | NaOH | Sigma-Aldrich | Cat. #: 415413 | Fecal phytate analysis |
| Chemical compound, drug | Calcium | Beckman Coulter | OSR6113 | Serum/urine biochemistry |
| Chemical compound, drug | Phosphate | Beckman Coulter | OSR6122 | Serum/urine biochemistry |
| Chemical compound, drug | BUN (Urea) | Beckman Coulter | OSR6134 | Serum/urine biochemistry |
| Chemical compound, drug | Creatinine | Beckman Coulter | OSR6178 | Serum/urine biochemistry |
| Chemical compound, drug | Magnesium | Beckman Coulter | OSR6189 | Serum/urine biochemistry |
| Chemical compound, drug | Urine protein | Beckman Coulter | OSR6170 | Urine biochemistry |
| Commercial assay or kit | Rat intact PTH | Immutopics | Cat. #: 60–2500 | ELISA kit |
| Commercial assay or kit | Human PTH | Abcam | Cat. #: ab230931 | ELISA kit |
| Commercial assay or kit | Rat soluble RANKL | Immundiagnostik | Cat. #: K1019 | ELISA kit |
| Commercial assay or kit | Rat osteoprotegerin | Alpco Immunoassay | Cat. #: 30–1020 | ELISA kit |
| Commercial assay or kit | Rat intact FGF23 | Elabscience | Cat. #: E-EL-R3031 | ELISA kit |
| Commercial assay or kit | Rat C-terminal FGF23 | Elabscience | Cat. #: E-EL-RB0377 | ELISA kit |
| Commercial assay or kit | Human FGF23 | Elabscience | Cat. #: E-EL-H1116 | ELISA kit |
| Commercial assay or kit | 25(OH)-Vitamin D | Abbott/USA | ARCHITECT 25-OH Vitamin D | Chemiluminescence microparticle immunoassay |
| Commercial assay or kit | 1,25-(OH)two vitamin D | DIAsoure | 1,25(OH)2-VIT, D-RIA-CT/Belgium | Radioimmunoassay |

*Continued on next page*

*Continued*

| Reagent type (species) or resource | Designation | Source or reference | Identifiers | Additional information |
|---|---|---|---|---|
| Commercial assay or kit | Hematoxylin solution | Thermo scientific | Cat. #: TA-125-MH | H&E staining |
| Commercial assay or kit | Eosin Y solution | Sigma-Aldrich | Cat. #: 318906 | H&E staining |
| Commercial assay or kit | Alizarin Red S | Sigma-Aldrich | Cat. #:130-22-3 | Kidney calcium staining |
| Commercial assay or kit | Von Kossa stain kit | Abcam | Cat. #:ab150687 | Kidney calcium staining |
| Commercial assay or kit | Trichrome stain kit | Abcam | Cat. #:ab150686 | Kidney fibrosis staining |
| Commercial assay or kit | TRAP kit | Sigma-Aldrich | Cat. #:387A | Bone ostaoclast |
| Commercial assay or kit | Oxalate (oxalic acid) Colorimetric assay kit | BioVision | Cat. #: K663 | ELISA kit |
| Commercial assay or kit | QIAamp Fast DNA Stool Mini Kit | Quigen | Cat. #:51604 | Stool DNA prepraration |
| Commercial assay or kit | THUNDERBIRD SYBR qPCR mix | Toyobo | Cat. #:QPS-101 | |
| Commercial assay or kit | TB Green Premix Ex Taq | TaKaRa | RR420L | |
| Commercial assay or kit | Antibody diluent with background-reducing components | DaKo | Cat. #:S3022 | IHC |
| Commercial assay or kit | Protein block, serum free | DaKo | Cat. #:X0909 | IHC |
| Commercial assay or kit | Liquid DAB + substrate chromogen system | DaKo | Cat. #:K3468 | IHC |
| Peptide, recombinant protein | Mouse RANKL | R and D Systems | Cat. #:462-TEC-010 | In vitro osteoclast |
| Peptide, recombinant protein | Mouse M-CSF | R and D Systems | Cat. #: 416 ML | In vitro osteoclast |
| Antibody | Mouse monoclonal β-actin | Santa Cruz | Cat. #: AC-15 | WB (1:1000) |
| Antibody | Rabbit polyclonal FGF23 | Bioss | Cat. #: bs-5768R | WB (1:200) IHC (1:100) |
| Antibody | Rabbit polyclonal αklotho | Novus Biologicals | Cat. #: NBP1-76511 | WB (1:500) |
| Antibody | Rabbit polyclonal SLC34A1/NaPi-2a | Abcam | Cat. #: ab151129 | WB (1:500) |
| Antibody | Rabbit polyclonal SLC34A2/NaPi-2b | LSBio | Cat. #: LS-C782282 | IHC (1:100) |
| Antibody | Rabbit polyclonal α-SMA | Abcam | Cat. #:ab5694 | IHC (1:100) |
| Antibody | Rabbit polyclonal TRPV6 | Novus Biologicals | Cat. #: NBP2-32372 | IHC (1:100) |
| Antibody | Rabbit polyclonal MINPP1 | LSBio | Cat. #: LS-C368910 | IHC (1:100) |
| Software, algorithm | GraphPad Prism 8.0 | GraphPad software | | Graph and statistics |

## Care and feeding of rats

Animal studies were approved by the Center for Animal Care and Use and were performed according to institutional ethics and safety guidelines (LCDI-2008–0019, LCDI-2010–0068, and LCDI-2018–0060). Three-week-old male (60–70 g) and female (50–55 g) Sprague–Dawley rats (Orient Bio, Seoul, Korea) were housed three per cage in a 12:12 hr light-dark cycle with ambient temperature maintained at $22 \pm 2°C$. The rats were fed AIN-93G (Dyets Inc, Bethlehem, PA, USA) for a 1 week adaptation period and were randomly assigned to control AIN-93G (0% phytate), AIN-93G with 1%, 3%, or 5% sodium phytate (P-8810, Sigma-Aldrich, St. Louis, MO, USA) or AIN-93G with 3% sodium phytate with an additional 0.5% $Ca^{2+}$ (n = 6–12 per group) diet for 12 weeks. All diets had equivalent percentages of carbohydrate, protein, fat, and minerals. Pathogen-free water and food were available ad libitum. Food intake and body weight were recorded weekly.

## Biochemical assays

The rats were fasted overnight every 2–3 weeks and anesthetized with isoflurane. Tail vein blood samples were stored at $4°C$ for 10–20 min and serum was frozen at $-80°C$. At the end of the experiment, the rats were fasted for 14–16 hr before receiving isoflurane. Abdominal aorta blood was obtained, and urine was collected over 24 hr. The kidneys and femora were dissected and stored at $-80°C$. Serum biochemical analysis was performed (Model AU-480; Beckman Coulter, Fullerton, CA, USA). Rat intact PTH (Immutopics, San Clemente, CA, USA), human intact PTH (Abcam), soluble RANKL (Immundiagnostik, Bensheim, Germany), osteoprotegerin (Alpco Immunoassay, Salem, NH, USA), rat intact and C-terminal FGF23 (Elabscience, Houston, TX, USA), and human intact FGF23 (Elabscience) were measured using enzyme-linked immunosorbent assay. The serum levels of 25-OH Vitamin D and $1,25(OH)_2$ vitamin D were determined by chemiluminescence microparticle immunoassay (Abbott, Abbott Park, IL) and radioimmunoassay (DIAsource, Louvain-la-Neuve, Belgium), respectively, in the biochemistry laboratory at Green Cross Labcell (Yongin, Korea). According to the manufacturers, the antibodies for 25-OH and $1,25(OH)_2$ vitamin D have cross-reactivity for vitamin D2 and D3 forms of 25-OH and $1,25(OH)_2$ vitamin D, respectively. In our assay, the serum levels of 25-OH vitamin D and $1,25(OH)_2$ vitamin D are the sum of both vitamin D2 and D3 levels. Creatinine clearance (Cl) was calculated as Cl (ml/min) = (urine [creatinine]×urine vol×body wt)/(serum [creatinine]×24 × 60). The fractional excretion of $Ca^{2+}$ or phosphate was calculated as the $\%FE_{Ca}$ or Pi = urine [$Ca^{2+}$ or phosphate] × serum [creatinine])/(urine [creatinine]×serum [$Ca^{2+}$ or phosphate]) × 100. TmP/GFR and $TmCa^{2+}$/GFR were calculated in accordance with the Walton and Bijvoet nomogram (*Walton and Bijvoet, 1975*) using a 24 hr urine sample and a serum sample obtained 16 hr after urine collection and fasting.

## Fecal mineral content analysis

Fecal samples were collected from rats after 2 or 4 weeks of either the control or phytate-supplemented diet. Rats were placed in grid cages over clean trays to collect samples. Food was removed from the cages during collection to avoid contamination. All samples were stored at $-80°C$. $Ca^{2+}$, magnesium, iron, and phosphate were quantified on an inductively coupled plasma optical emission spectrometer (Optima 5300 DV; PerkinElmer, Waltham, MA, USA). Fecal samples were dried at $60°C$ for 48 hr, and 1.0 g was heated at $560°C$ for 4 hr. After cooling to room temperature, 1 mL of 35.5% $HNO_3$ was added and samples were heated at $560°C$ for an additional 2 hr. Samples were then exposed to 4 mL 17.5% HCl at room temperature for 2 hr before dilution in ultra-pure water. Fecal phytate was sequentially extracted with a HCl-HF acid solution, followed by an alkaline extraction (NaOH and EDTA), as previously described (*Ray et al., 2012*). The phytate content of fecal extracts was quantified by high-performance ion chromatography (ICS-3000; Dionex).

## Kidney histopathology

Hematoxylin and eosin (HE)-stained kidney sections (n = 64) from eight male and eight female rats were blindly examined by one author (CJB). $Ca^{2+}$ was stained with Alizarin Red S and von Kossa. Sections obtained from females were stained with Periodic acid–Schiff. Parameters were scored semiquantitatively on the basis of previous work (*Lalioti et al., 2006*). The measured parameters included the presence and severity of mineralization, inflammation, tubular casts, tubular ectasia, interstitial fibrosis, tubular degeneration/regeneration, and glomerular hypertrophy (0 = negative or

within normal limits, 1 = minimal, 2 = mild, 3 = moderate, 4 = marked, and 5 = severe). Individual scores were added together for each kidney, with a maximum score of 40.

## In vivo magnetic resonance imaging (MRI)

We used a 9.4 T animal MRI scanner (Biospec 94/20 USR; Bruker Biospin, Ettlingen, Germany) with a transmit-only volume coil for excitation and phased-array four-channel surface coil for signal reception (Bruker Biospin). Animals were anesthetized using isoflurane. MRI scans were performed using the following respiratory-gated, fat-suppressed gradient echo sequence: TR/TE, 5000/28.5 ms; FA, 180˚; FOV, $70 \times 60$ mm$^2$; matrix size, $384 \times 384$; TH, 1.0 mm; 27 slices (no gap); and NEX, 4. Axial images were acquired for anatomic reproducibility and histopathological correlation.

## Bone mineral density (BMD)

The right femora samples were harvested from eight mice per group after 12 weeks on either control or phytate-supplemented diet. Ex vivo BMD, bone mineral content, and femoral lengths were measured by peripheral dual-energy X-ray absorptiometry (pDEXA) (Sabre; Norland Medical Systems Corp., Fort Atkinson, WI, USA) with pDEXA Sabre software (version 3.9.4; Norland). Scans of femora ($4 \times 4$ cm$^2$) were performed at 3 mm/s and resolution of $0.2 \times 0.2$ mm. BMD was computed using dual X-ray beam attenuation (28 and 48 KeV), and BMD histogram averaging widths were 0.02 g/cm$^2$. In vitro precision, expressed as the coefficient of variation (CV = 100 × standard deviation/ mean), was calculated by the BMD of a phantom with a nominal density of 0.929 g/cm$^2$.

## μ-computed tomography (μ-CT) analysis

The femora were isolated from soft tissue, fixed overnight in 10% formalin, and analyzed by high-resolution μ-CT (SkyScan1172 in vivo CT; Bruker). Imaging (NRecon ver. 1.6; Bruker), data analysis (CTAn ver. 1.11; Bruker), and three-dimensional visualization (μ-CTVol ver. 2.2) were performed on 500 slices per scan at 70 kVp, 141 μA, and 11.55 μm pixel$^{-1}$. Cross-sectional images were obtained for three-dimensional histomorphometry. A 1.2 mm sample of metaphyseal secondary spongiosa was imaged. Cross-sectional images (500 images per specimen) of distal femora were used for two-dimensional and three-dimensional morphometry.

## Bone histomorphometry

The right femora were fixed in 10% formalin for 48 hr, decalcified with 8% hydrochloric acid/formic acid for 2 weeks, and embedded in paraffin. Longitudinally oriented 2.5 μm thick bone sections (including metaphyses and diaphyses) were processed for HE and Masson's trichrome stains. The left femora were fixed in formalin for 48 hr, decalcified in 10% EDTA (pH 7.0) for 4–5 weeks at 4˚C, and embedded in paraffin for tartrate-resistant acid phosphatase staining. Sections were viewed using a Zeiss Axio Imager Z1 microscope (Carl Zeiss, Oberkochen, Germany). Histomorphometric measurements were made at the center of the metaphyses. Images were captured with 3DHISTECH (Budapest, Hungary). Bone sections (2.5 μm thick) were stained with Masson's trichrome for osteoids.

## Rat bone marrow osteoclasts

Bone marrow cells were isolated using a modified method (*Kobayashi et al., 2000*). Tibiae and femora were isolated from rats after 12 weeks on either control or phytate-supplemented diet. Bone ends were removed and the marrow was flushed by slowly injecting serum-free alpha minimal essential medium (αMEM; Invitrogen, Carlsbad, CA, USA) through a sterile 25-gauge needle into one end of the bone. Marrow cells were collected into tubes, washed twice with αMEM, and cultured in αMEM with 10% heat-inactivated fetal bovine serum (Invitrogen). Cells were cultured in 24-well plates ($1 \times 10^5$/ well) with 30 ng/mL of macrophage colony-stimulating factor (M-CSF) for 72 hr. Differentiation of M-CSF-dependent bone marrow osteoclast precursors was induced with 30 ng/mL receptor activator of nuclear factor kappa-B ligand (RANKL) (R and D Systems, Minneapolis, MN, USA) and 30 ng/mL M-CSF for 4 days. Differentiated osteoclasts with more than three nuclei were identified as tartrate-resistant acid phosphatase-positive (Leukocyte Acid Phosphatase Assay, Sigma).

## Western blotting and immunohistochemistry (IHC)

The primary antibodies used for Western blotting or immunohistochemistry were monoclonal β-actin (AC-15, 1:1000; Santa Cruz), FGF23 (bs-5768R, 1:200; Bioss), αKlotho (NBP1-76511, 1:500; Novus Biologicals), NHERF1 (A-7, 1:200; Santa Cruz), NaPi-2a (ab182099, 1:500; Abcam), and NaPi-2a (ab151129, 1:500; Abcam), as described previously (*Kang et al., 2017*). Antigen retrieval for IHC was performed in Tris/EDTA (pH 9.0) for 2–3 min. IHC was performed with the following antibodies at 25℃ for 2 hr: α-SMA (ab5694, 1:100; Abcam), NaPi-2b (S ab182099, 1:100; Abcam), TMPV6 (NBP2-32372, 1:100; Novus Biologicals), MINPP1 (LS-C368910, 1: 100; LSBio), or FGF23 (bs-5768R, 1:200; Bioss), Horseradish peroxidase-conjugated secondary antibodies were detected with 3,3,-dia-mino-benzidine tetrahydrochloride substrate (Dako, Glostrup, Denmark). All sections were counter-stained with hematoxylin (Themo Scientific).

## qRT-PCR analyses

Total RNA was extracted from whole kidney lysates with Trizol (Invitrogen). cDNA synthesis and real-time RT-PCR were performed using TB Green Premix Ex Taq cDNA Synthesis kit (TaKaRa Bio, Tokyo, Japan) with 1 μg total RNA plus random hexamers in BioRad CFX384. All expression values were normalized to the *cyclophilin A* mRNA level. Primer sequences are listed in *Supplementary file 6*.

## Phytate pilot study in humans

A noncomparative pilot study was approved by the Institutional Review Board of Dongguk University Ilsan Hospital (IRB No. 2014–117). All procedures were performed in accordance with the Declaration of Helsinki. The study was registered with Clinical Research Information Service (registration number KCT0003144) in accordance with the WHO International Clinical Trials Registry Platform. This study aimed to evaluate the effects of phytate intake on mineral metabolism in humans between January and February 2015. All participants provided written informed consent. In total, six premenopausal women aged between 23 and 34 years with a body mass index between 17.7 and 25.8 kg/$m^2$ were recruited. The exclusion criteria were as follows: bone and mineral metabolism disorders (e. g., hyperparathyroidism, hypoparathyroidism, osteomalacia); previous surgical history which could affect bone and mineral metabolism (e.g., parathyroidectomy, thyroidectomy, and gastrectomy); use of any medication that could affect bone and mineral metabolism (e.g., estrogen, bisphosphonates, selective estrogen receptor modulators, and calcitonin); use of $Ca^{2+}$ and vitamin D supplementation within the past 3 months; use of corticosteroids within the past 1 month; acute or chronic renal failure; exhibited abnormal liver function; or people who were pregnant or lactating. All participants completed a 12-day trial consisting of three successive diet cycles containing: (1) polished rice (low phytate content, 0.35%); (2) brown rice (medium phytate content, 1.07%); and (3) brown rice mixed with rice bran (high phytate content, 2.17%). Each diet cycle lasted for 4 days. During each day of the study, all participants were provided with breakfast, lunch, dinner, and a snack. Every meal was packed in a lunch box and delivered each day, and participants were not allowed to consume any other food during the trial except for the food provided by study investigators. All meals were identical across the three diet cycles except for the phytate content. Water consumption was allowed ad libitum. No medications or supplements that affected mineral metabolism were allowed during the course of the study. A blood sample was collected from each participant at the beginning of the trial and at the end of each diet cycle. Urine samples of 24 hr were collected before the trial and during the fourth day of each diet cycle.

## Microbiota analysis

Genomic DNA was isolated from frozen stool samples using the QIAamp fast DNA stool kit (Qiagen, Valencia, CA) according to the manufacturer's instructions. Levels of 16S rRNA gene expression of each bacterial phylum were quantified using THUNDERBIRD SYBR qPCR mix (Toyobo, Osaka, Japan) on a CFX Connect real-time PCR detection system (Bio-Rad, Hercules, CA).

## Statistical analysis

Sample sizes for all experiments were based on preliminary results and previous experience of conducting related experiments. Power calculations were not used to determine sample sizes. Animals for each group of experiments were randomly assigned. No animals were excluded from statistical

analysis, and investigators were blinded to the study condition when scoring histopathological samples. Unless noted otherwise, all data are presented as the mean ± standard deviation (SD). Statistical comparisons of groups were performed by either unpaired Student's *t* test or one-way ANOVA with Tukey's post hoc multiple comparison test. All data were subjected to D'Agostino and Pearson (n > 8) or Shapiro–Wilk (n < 8) normality test before analysis. Those groups that failed the normality test (p<0.05) were subjected to an outlier test (ROUT; Q = 1%), as recommended to determine whether the outlier was responsible for the failure of the normality test. If the exclusion of outliers led to passing the normality test and altered the statistical results, the exclusion was made; if it did not alter the statistical outcome, no data were excluded from that group. The Kruskal–Wallis test was used for non-parametric statistical analysis of data that did not have a normal distribution. We performed data analysis and prepared graphs using GraphPad Prism 8.0 (GraphPad Software Inc, San Diego, CA).

## Acknowledgements

The authors thank Dr S Dhe-Paganon and S Shoelson at Harvard Medical School for their valuable suggestions regarding the manuscript.

## Additional information

### Funding

| Funder | Grant reference number | Author |
| --- | --- | --- |
| National Research Foundation of Korea | 2017R1D1A1B03031094 | Ok-Hee Kim |
| National Research Foundation of Korea | 2019R1A2C2008130 | Byung-Chul Oh |
| Ministry of Food and Drug Safety | 14182MFDS460 | Byung-Chul Oh |
| Korea Health Industry Development Institute | HI14C1135 | Byung-Chul Oh |
| Ministry of Food and Drug Safety | 16181MFDS381 | Byung-Chul Oh |

The funders had no role in study design, data collection and interpretation, or the decision to submit the work for publication.

### Author contributions

Ok-Hee Kim, Conceptualization, Data curation, Funding acquisition, Writing - original draft, Writing - review and editing; Carmen J Booth, Young Jae Lee, Cheol Soon Lee, Cheolsoo Choi, Yun Jae Jung, Sun Wook Cho, Data curation, Formal analysis; Han Seok Choi, Data curation, Formal analysis, Investigation; Jinwook Lee, Investigation; Jinku Kang, June Hur, Formal analysis, Investigation; Woo Jin Jung, Yun-Shin Jung, Hyung Jin Choi, Hyeonjin Kim, Joong-Hyuck Auh, Data curation; Jung-Wan Kim, Ji-Young Cha, Seung-Soon Im, Formal analysis; Jun-Young Yang, Resources, Formal analysis; Dae Ho Lee, Conceptualization, Formal analysis, Writing - review and editing; Young-Bum Kim, Kyong Soo Park, Conceptualization, Formal analysis; Young Joo Park, Conceptualization, Formal analysis, Writing - original draft, Writing - review and editing; Byung-Chul Oh, Conceptualization, Formal analysis, Funding acquisition, Writing - original draft, Project administration, Writing - review and editing

### Author ORCIDs

Byung-Chul Oh (ID) https://orcid.org/0000-0002-4277-7083

### Ethics

Clinical trial registration A noncomparative pilot study was approved by the Institutional Review Board of Dongguk University Ilsan Hospital (IRB No. 2014-117). All procedures were performed in

accordance with the Declaration of Helsinki. The study was registered with Clinical Research Information Service (registration number KCT0003705; https://cris.nih.go.kr/cris/en/search/search_result_st01.jsp?seq=13397) in accordance with the WHO International Clinical Trials Registry Platform. This study aimed to evaluate the effects of phytate intake on mineral metabolism in humans between January and February 2015.

Human subjects: A noncomparative pilot study was approved by the Institutional Review Board of Dongguk University Ilsan Hospital (IRB No. 2014-117). All procedures were performed in accordance with the Declaration of Helsinki. The study was registered with Clinical Research Information Service (registration number KCT0003705; https://cris.nih.go.kr/cris/en/search/search_result_st01.jsp?seq=13397) in accordance with the WHO International Clinical Trials Registry Platform. This study aimed to evaluate the effects of phytate intake on mineral metabolism in humans between January and February 2015. All participants provided written informed consent. In total, six premenopausal women aged between 23 and 34 years with a body mass index between 17.7 and 25.8 kg/m2 were recruited. The exclusion criteria were as follows: bone and mineral metabolism disorders (e.g., hyperparathyroidism, hypoparathyroidism, osteomalacia), previous surgical history which could affect bone and mineral metabolism (e.g., parathyroidectomy, thyroidectomy, gastrectomy), use of any medication that could affect bone and mineral metabolism (e.g., estrogen, bisphosphonates, selective estrogen receptor modulators, calcitonin), use of Ca2+ and vitamin D supplementation within the past 3 months, use of corticosteroids within the past 1 month, or acute or chronic renal failure; exhibited abnormal liver function; or were pregnant or lactating.

Animal experimentation: Animal studies were approved by the Center for Animal Care and Use and were performed according to institutional ethics and safety guidelines (LCDI-2008-0019, LCDI-2010-0068, and LCDI-2018-0060).

### Decision letter and Author response
Decision letter https://doi.org/10.7554/eLife.52709.sa1
Author response https://doi.org/10.7554/eLife.52709.sa2

## Additional files

### Supplementary files
• Supplementary file 1. Composition of the AIN-93G rodent diets for experimental studies with rats.
• Supplementary file 2. The nutrient compositions for three different diets.
• Supplementary file 3. Biochemical properties during three consecutive diets.
• Supplementary file 4. Phylum- and class-specific primers for real-time PCR.
• Supplementary file 5. Biochemical and structural properties of mammalin, gut microbial, and bacterial phytases.
• Supplementary file 6. The qRT-PCR primer lists.
• Transparent reporting form

### Data availability
Our primer lists are provided in Supplementary files 4 and 6, and the summary of in a dietary study of humans is provided as Supplementary file 2.

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
