## [Decision Letter]

**Acceptance summary:**

The paper defines mechanistically, using young growing rats, the dietary profile (high phytate and low calcium content) that promotes intestinal phosphate overload and phytate-induced renal and bone disease. This dietary composition induces reductions in bone mass, osteoclastogenesis, phosphate wasting, secondary hyperparathyroidism, and nephrocalcinosis. The high dietary phytate effects are counteracted with high calcium intake. In a small 12-day human pilot study of 6 premenopausal women, the authors noted decreases in serum phosphate and urinary calcium levels with progressive increases in dietary phytate. Overall, the study has translational implications for nutritional rickets and osteomalacia; for growing children and adolescents consuming diets high in phytate (cereal-based foods) and low in calcium; and for individuals with kidney and metabolic bone diseases.

**Decision letter after peer review:**

Thank you for resubmitting your work entitled "High-phytate/low-calcium diet is a risk factor for crystal nephropathies, renal phosphate wasting, and bone loss" for further consideration by *eLife*. Your revised article has been evaluated by Clifford Rosen as the Senior Editor and a Reviewing Editor.

The manuscript is considered potentially acceptable, but there are some issues that need to be addressed before final acceptance, as outlined below:

It would be important to clarify Figure 1 legends and the statistical results which are somewhat confusing.

Please discuss the rationale for using 3 week rats and the differences by sex which seem somewhat unusual.

Clarify the methods and measurements of the vitamin D metabolite.

Are there other measures of renal function such as inulin clearance?

These responses should be relative straightforward as noted below:

Reviewer #1:

Kim et al. conduct in their article "High-phytate/low-calcium diet is a risk factor for crystal nephropathies, renal phosphate 1 wasting, and bone loss" a careful analysis of diets with different phytate and calcium contents on renal function and mineral and bone metabolism. They translate their findings in rats to humans. However, the interpretation of their data would benefit from some clarification: they need to separate bioaccessibility from bioavailability of calcium and phosphate. The former is measured with the ratio of soluble vs. insoluble calcium and phosphate in the feces, the latter is reflected by uptake of radioactive Ca and Pi tracers, which in part reflects earlier work of the authors. Their findings are not novel per se, but they present nice studies to evaluate the mechanism of how calcium supplements affect bioaccessibility and bioavailability of phytate-phosphate, which would benefit from conceptually separating luminal and blood/extracellular compartments when discussing effects of Ca and Pi on renal function and mineral and bone metabolism. I am not sure, whether I would arrive at the same conclusion: could it be that phytate is cleaved in the gut and free phosphate reduces bioaccessibility and thereby bioavailability of calcium? Resulting hypocalcemia could explain secondary hyperparathyroidism, bone loss and phosphaturia, which in turn causes nephrocalcinosis. Addition of extra calcium to the diet prevents secondary hyperparathyroidism and other consequences.

Reviewer #2:

Kim et al. present in well-written manuscript extensive data indicating that a high-phytate/low-calcium diet leads to nephrocalcinosis, hyperparathyroidism induced by reduced calcium absorption leading to hypocalcemia. The increase in PTH enhances phosphate excretion.

1) Renal function is impaired in the group on the high-phytate diet; it would be nice to provide additional measures of kidney function, for example, through the assessment of inulin clearance.

2) Did the author measure oxalate levels in blood? Could an increase in intestinal oxalate absorption contribute to the development of nephrocalcinosis and CKD?

Reviewer #3:

Thank you for the opportunity of reviewing the manuscript by Kim et al., where the authors showed that a diet high in phytate and low in calcium leads to mineral and bone disorder and nephropathy in rats. I hope the authors find helpful my suggestions and comments:

Methods

– Why 3-wk old rats? These rats are still in a period of bone and mass accrual. I ask this because this may limit translation of the result into human adults.

– What is the total amount of Calcium and phosphorus in the different diets? I would suggest adding this as supplementary information.

– For the metabolic studies, how long were these? 24h?

– For the human study, do the authors think the background diet may affect the results? There were only 6 females included and the results are highly variable. Please include the diets in terms of macro and micronutrients.

Results

– Figure 1 (text and figure)- if the animals are consuming similar amounts of food, what is the mechanism for reductions in fat-free mass? There is no discussion about this.

– Figure 3 – why did the bone assessment was done only 0 vs. 5% phytate diets? Are the RANKL values in males or females or both? Same thing with Bone and BMC TRAP?

– Figure 4 (figure and text)- why were only females studied? what was the overall diet composition? why were feces not collected to see phytate excretion? What was the calcium intake?

– Figure 5 (text and figure)- It is hard to read the panels.

– Figure 6- what tissue/specimen are panels D-F?

– Figure 7: I like the model, but key factors are missing? How is phytate metabolized and components absorbed? Is there a change in phytase from the gut microbiota? I may be biased, but it is my understanding that phytase is not present in the small intestine in mammals. Then, how is the digestibility change? Also, FGF23 is missing from the figure.

– Why is urinary pH changed?

– The main hypothesis is related to the digestion of phytate and absorption of minerals, but the intestine was not assessed. Did the authors collected small intestine? If so, I would recommend assessing changes in calcium and phosphorus transporters. Also, how exactly is phytate metabolized?

Discussion

– Members of the gut microbiota express phytase, there is no discussion on this and how the different types of diets may impact this.

Supplementary figures

– Figure 4—figure supplement 1: Is there a difference in FGF23?

---

## [Author Response]

The manuscript is considered potentially acceptable, but there are some issues that need to be addressed before final acceptance, as outlined below:It would be important to clarify Figure 1 legends and the statistical results which are somewhat confusing.

Following discussion with an expert in statistics, we performed data analysis and prepared graphs using GraphPad Prism 8.0. We have described the statistical methods in the Materials and methods section in the revised manuscript as follows: “All data are presented as the mean ± standard deviation (SD). Statistical comparisons of groups were performed by either unpaired Student’s *t* test or one-way ANOVA with Tukey’s post hoc multiple comparison test. All data were subjected to D’Agostino and Pearson (*n* > 8) or Shapiro–Wilk (*n* < 8) normality test before analysis. Those groups that failed the normality test (*P* < 0.05) were subjected to an outlier test (ROUT; Q = 1%), as recommended to determine whether the outlier was responsible for the failure of the normality test. If the exclusion of outliers led to passing the normality test and altered the statistical results, the exclusion was made; if it did not alter the statistical outcome, no data were excluded from that group. The Kruskal–Wallis test was used for non-parametric statistical analysis of data that did not have a normal distribution.”

Accordingly, the statistical significance has been corrected in Figure 1G, H and L, Figure 2B, C and H, Figure 3C, Figure 6A, B and C, as well as Figure 1–figure supplement 1E, Figure 1—figure supplement 2D, E, and F), and Figure 5—figure supplement 3A in the revised manuscript. In Figure 1L, One sample of 3% phytate in male rats was out of range according to ROUT (Q = 1%) but it did not alter the statistical outcome and no data were excluded from that group. The type of statistical test used for the analysis in each graph has been stated in the corresponding figure legend. A detailed description of the group comparison statistical test has been added to the Materials and methods section.

For a better description of the data, we elaborated on the experimental design in Figure 1A and molar ratio of Ca^2+^ to phytate in Figure 1A as follows:

“(A) Experimental design. After a 1-week period of adaptation with control AIN93G diet, we fed 4-week-old male and female Sprague–Dawley rats with AIN-93G diets supplemented with 0% (control), 1%, 3%, or 5% phytate for 12 weeks. […] All data are presented as the mean ± standard deviation (SD) for each group (*n* = 8 – 12 per group). **P* < 0.05, ***P* < 0.01, ****P* < 0.001 compared with controls.”

Please discuss the rationale for using 3 week rats and the differences by sex which seem somewhat unusual.

We obtained 3-week-old rats and started the phytate diet studies at 4 weeks of age after a 1-week adaptation period with the control AIN-93G diet after weaning, as described in Figure 1A – and the Materials and methods. Four weeks of age is an appropriate time to start a diet-based study. Although longitudinal growth ceases, the rats maintain growth plates for life, and Roach (Roach et al.) reported that it was appropriate to start a study at weaning to examine the effects of diet on bone in a rat model.

We hypothesized that a high-phytate/low-Ca^2+^ diet is more harmful to young rats that are growing actively because a higher Ca^2+^ content is required for bone mass accrual and the growing skeleton. Therefore, we used 4-week-old rats in these experiments. We think that our results might be translated to humans in that a high-phytate/low-Ca^2+^ diet could have a greater effect on children and adolescents than adults.

As the reviewer noted, it is well known that there are sex-related differences in bone metabolism in rats and humans (Czerniak, 2001; Yan et al., 2002), although the reason for these differences has not been elucidated. In our experiments, female rats were slightly more likely to develop nephrocalcinosis, but the sex-related differences in the effects of the phytate diets on bone metabolism were not significant. Currently, we cannot explain the greater sensitivity of female rats, and further research is needed.

Clarify the methods and measurements of the vitamin D metabolite.

The serum levels of 25-OH Vitamin D and 1,25(OH)_2_ vitamin D were determined by chemiluminescence microparticle immunoassay (Abbott, Abbott Park, IL) and radioimmunoassay (DIAsource, Louvain-la-Neuve, Belgium), respectively. According to the manufacturers, the antibodies for 25-OH and 1,25(OH)_2_ vitamin D have cross-reactivity for vitamin D2 and D3 forms of 25-OH and 1,25(OH)_2_ vitamin D, respectively. In our assay, the serum levels of 25-OH vitamin D and 1,25(OH)_2_ vitamin D are the sum of both vitamin D2 and D3 levels. Therefore, we corrected the names of vitamin D metabolites as 25-(OH)D and 1,25(OH)_2_D in the revised manuscript. Thank you for the reviewer’s comment.

Are there other measures of renal function such as inulin clearance?

As the reviewer’s comment, inulin, a small polysaccharide molecule, is now considered the gold standard for measuring glomerular filtration rate (GFR) in research. Inulin clearance is determined by measuring the rate at which inulin is cleared for blood plasma. However, additional experiments are required to measure inulin clearance because live animals are required. So, we cannot provide inulin clearance in our experiments.

To address the reviewer’s comments, we measured 24-hour urinary protein/Cr ratio as a marker of renal damage although rats fed the HP-LCa^2+^ diet had impaired renal function, such as hypophosphatemia, high serum BUN level, high BUN/Cr ratio, and nephrocalcinosis, and the results indicated that rats fed the HP-LCa^2+^ diet had a markedly increased 24-hour urinary protein/creatinine ratio compared to controls.

We have briefly described these results in the revised manuscript as follows: “Furthermore, rats fed the HP-LCa^2+^ diet had a markedly increased 24-hour urinary protein/Cr ratio compared to controls, suggesting that a high-phytate diet contributes to the development of proteinuria, a sign of renal damage (Figure 5—figure supplement 1H).

These responses should be relative straightforward as noted below:Reviewer #1:Kim et al. conduct in their article "High-phytate/low-calcium diet is a risk factor for crystal nephropathies, renal phosphate 1 wasting, and bone loss" a careful analysis of diets with different phytate and calcium contents on renal function and mineral and bone metabolism. […] Addition of extra calcium to the diet prevents secondary hyperparathyroidism and other consequences.

In this study, we used high-phytate/low-Ca^2+^ and high-phytate/high-Ca^2+^ diets. In the high-phytate/low-Ca^2+^ diets, the majority of phytate might exist as a form of Ca^2+^-free phytate and the amount would increase with the percentage of phytate in diets with the same Ca^2+^ content.

Ca^2+^-free phytate is hydrolyzed to inositol phosphate and free phosphate in the gut (increased bioaccessibility), which may result in increased absorption of phosphate into the blood (increased bioavailability) and cause renal phosphate wasting (hyperphosphaturia). Most Ca^2+^ in the high-phytate/low-Ca^2+^ diets complexed with phytate to form Ca^2+^-phytate, and the amount of free Ca^2+^ decreased gradually with the increment in phytates in the diet, reducing the bioaccessibility of Ca^2+^. This, in turn, could result in a decrement in the absorption of Ca^2+^ into the blood (decreased bioavailability), contributing to the development of hypocalcemia, resulting in hypocalcemia, as the reviewer commented. The hypophosphatemia combined with the high-phytate/low-Ca^2+^ diet could explain the secondary hyperparathyroidism, bone loss, and phosphaturia, which in turn causes nephrocalcinosis.

By adding extra calcium to the high-phytate/low-Ca^2+^ diet (giving the high-phytate/high-Ca^2+^ diet), the increased Ca^2+^ could increase the Ca^2+^-phytate and decrease Ca^2+^ free-phytate. This might decrease the absorption of phosphate. Under these conditions, the amount of fecal Ca^2+^-phytate should increase, as shown by the ICP-MS results for fecal Ca^2+^ and phosphate and the fecal phytate content determined by HPIC. The increased amount of calcium could also increase the amounts of free Ca^2+^ and Ca^2+^-phytate complex. Therefore, we postulated that the deleterious effects of a high phytate diet could be reduced by adding extra calcium to the diet.

As the reviewer suggested, we have described this in the Discussion as follows: “The resulting hypocalcemia and hypophosphatemia caused by the high-phytate/low-Ca^2+^ diet could explain the secondary hyperparathyroidism, bone loss and phosphaturia, which in turn causes nephrocalcinosis.”

Reviewer #2:Kim et al. present in well-written manuscript extensive data indicating that a high-phytate/low-calcium diet leads to nephrocalcinosis, hyperparathyroidism induced by reduced calcium absorption leading to hypocalcemia. The increase in PTH enhances phosphate excretion.1) Renal function is impaired in the group on the high-phytate diet; it would be nice to provide additional measures of kidney function, for example, through the assessment of inulin clearance.

As the reviewer’s comment, inulin, a small polysaccharide molecule, is now considered the gold standard for measuring glomerular filtration rate (GFR) in research. Inulin clearance is determined by measuring the rate at which inulin is cleared for blood plasma. However, additional experiments are required to measure inulin clearance because live animals are required. So, we cannot provide inulin clearance in our experiments.

To address the reviewer’s comments, we measured 24-hour urinary protein/Cr ratio as a marker of renal damage although rats fed the HP-LCa^2+^ diet had impaired renal function, such as hypophosphatemia, high serum BUN level, high BUN/Cr ratio, and nephrocalcinosis, and the results indicated that rats fed the HP-LCa^2+^ diet had a markedly increased 24-hour urinary protein/creatinine ratio compared to controls.

We have briefly described these results in the revised manuscript as follows: “Furthermore, rats fed the HP-LCa^2+^ diet had a markedly increased 24-hour urinary protein/Cr ratio compared to controls, suggesting that a high-phytate diet contributes to the development of proteinuria, a sign of renal damage (Figure 5—figure supplement 1H).

2) Did the author measure oxalate levels in blood? Could an increase in intestinal oxalate absorption contribute to the development of nephrocalcinosis and CKD?

To address the reviewer’s comments, we measured the serum levels of oxalate in rats fed control, HP-LCa^2+^, and HP-HCa^2+^ diets. The results indicated no differences in serum oxalate levels between diet groups. Indeed, our renal histology such as Alizarin Red S and von Kossa staining suggested that the high-phytate-induced nephrocalcinosis was mainly associated with medullary calcium deposits.

We have briefly described these results in the revised manuscript as follows: “However, we found no differences in serum levels of oxalate between diet groups (Figure 5—figure supplement 3G), suggesting that phytate-induced nephrocalcinosis was mainly with medullary calcium deposits, consistent with renal histology, such as Alizarin Red S and von Kossa staining.”

Reviewer #3:Thank you for the opportunity of reviewing the manuscript by Kim et al., where the authors showed that a diet high in phytate and low in calcium leads to mineral and bone disorder and nephropathy in rats. I hope the authors find helpful my suggestions and comments:Methods– Why 3-wk old rats? These rats are still in a period of bone and mass accrual. I ask this because this may limit translation of the result into human adults.

In this study, we used 3-week-old rats but started phytate diet studies at 4 weeks old after a 1-week adaptation period with the control AIN-93G diet, as described in Figure 1A and the Materials and methods section. Even if the rats are weaned at 3 weeks old, this is an appropriate time to start a diet-based study. Four weeks of age is an appropriate time to start a diet-based study. While longitudinal growth ceases, they maintain growth plates for life, so starting a diet study at weaning to examine the effects of diet on bone in a rat model is appropriate (Roach et al., 2003).

In this study, we hypothesized that a high-phytate/low-Ca^2+^ diet is more harmful to young rats that require higher Ca^2+^ content for bone mass accrual and the growing skeleton. Therefore, we used the 4-week-old rats in these experiments. We think that our results might be translated into humans in that a high-phytate/low-Ca^2+^ diet could have a greater effect on children and adolescents than adults.

As the reviewer pointed out, it has been known there exist sex-related differences in bone metabolism in rats and humans (Czerniak, 2001; Yan et al., 2002). The reason for those differences has not been elucidated yet. In our experiments, although female rats showed slightly greater sensitivity for the development of nephrocalcinosis, the sex-related differences in the effects of phytate diets on bone metabolism were not significant. At this moment, we cannot explain the greater sensitivity for the development of nephrocalcinosis on female rats, and further researches should be needed.

– What is the total amount of Calcium and phosphorus in the different diets? I would suggest adding this as supplementary information.

We have added the compositions of the different diets as well as total amounts of Ca^2+^ and phosphate in Supplementary file 1 in the revised manuscript.

– For the metabolic studies, how long were these? 24h?

As described previously in the Materials and methods section and the figures, we collected urine for 24 hours and serum samples were obtained after 16 hours of urine collection and fasting for metabolic studies.

– For the human study, do the authors think the background diet may affect the results? there were only 6 females included and the results are highly variable. Please include the diets in terms of macro and micronutrients.

Based on the results in rodents, we have conducted a phytate clinical pilot study. As the reviewer pointed out, this study was performed only in six human female subjects, which is prone to variability. Therefore we tried to reduce the variability by qualifying the diet composition. We provided identical meals across the three diet cycles except for the phytate content contained in rice. That is, the only difference among the diets was the composition of rice provided; white rice, brown rice or brown rice with rice bran. We calculated the composition of the diets with an expert on clinical nutrition before the clinical study. The nutrient contents in terms of total energy, carbohydrate, protein, fat, and micronutrients were analyzed using the software CAN-Pro 5.0 (Computer-Aided Nutritional analysis program, version 5; The Korea Nutrition Society, 2015), and the calculated compositions of the nutrients were not different. The nutrient compositions for the three different diets were presented in Supplementary file 2.

Results– Figure 1 (text and figure)- if the animals are consuming similar amounts of food, what is the mechanism for reductions in fat-free mass? There is no discussion about this.

We found that rats fed control and phytate diets consumed similar amounts of the diets. However, rats fed the high-phytate diets showed a reduction in fat-free lean mass, mainly muscle mass. At present, we cannot explain the exact mechanism for the reduction of muscle mass in rats fed the high-phytate diets. One possible mechanism is increased muscle catabolism in rats fed the high-phytate diets, which is supporting by the high blood urea nitrogen (BUN) levels and a high BUN/Cr ratio as BUN is a serum byproduct/waste product of protein metabolism (Haines et al., 2019). A reduced bioavailable amino acid pool for muscle protein synthesis could be another potential mechanism because phytate is a well-known inhibitor of digestive enzymes, such as proteases. But the exact mechanism remains to be elucidated yet.

We briefly discussed this issue in the Results section in the revised manuscript as follows: “Elevated BUN/Cr ratio may provide a biochemical signature of increased muscle catabolism associated with decreased fat-free lean mass in rats fed a high-phytate diet (Haines et al., 2019).”

– Figure 3 – why did the bone assessment was done only 0 vs. 5% phytate diets? Are the RANKL values in males or females or both? Same thing with Bone and BMC TRAP?

We performed peripheral dual-energy x-ray absorptiometry (pDEXA) of the excised femora to determine the effects of phytate feeding on bone mass in male and female rats after 12 weeks of feeding with the experimental diets (0, 1, 3, and 5% phytate). As shown in Figure 3A, phytate-fed rats had significantly decreased bone mineral density in a phytate dose-dependent manner compared with the controls. Based on these results, we compared the bone assessments shown in Figure 3 in the 5% phytate diet and control in both male and female rats. This demonstrated that both sexes were affected. Next, we performed bone histomorphology (e.g., TRAP staining of bone sections and BMC TRAP staining) and measured the serum RANKL levels in female rats because the difference was more evident in female rats.

– Figure 4 (figure and text)- why were only females studied? what was the overall diet composition? why were feces not collected to see phytate excretion? What was the calcium intake?

Analyses of the impact of phytate feeding on calcium and phosphate homeostasis in rats indicated that female rats were slightly more susceptible to renal dysfunction and kidney stones compared to male rats. Therefore, we performed a clinical pilot study of phytate in human female subjects.

As answered to the prior comment, we analyzed the compositions of three different diets with an expert on clinical nutrition, the calculated compositions of the nutrients were not different, and have added the nutrient compositions of the three different diets in the Supplementary file 2.

Unfortunately, we did not collect feces to analyze fecal phytate but did collect 24-hour urine and serum to analyze the effects of phytate feeding on calcium and phosphate homeostasis in humans. Consistent with the results of rodent studies, we found that high-phytate diets dysregulated mineral and phosphate metabolism in humans, especially decreasing urinary Ca^2+^/Cr clearance and increasing tubular Ca^2+^ reabsorption as well as increasing serum levels of intact PTH.

– Figure 5 (text and figure)- It is hard to read the panels.

We have rewritten the legend for Figure 5 as follows:

“Figure 5. Ca^2+^ supplementation protects phytate-fed rats from renal phosphate wasting disorders, crystal nephropathies, and bone loss via indigestible Ca^2+^-phytate salts. […] All data are presented as the mean ± SD of each group (*n* = 4 – 6 per group). **P* < 0.05, ***P* < 0.01, ****P* < 0.001, compared with controls. ^#^*P* < 0.05, ^##^*P* < 0.01, ^###^*P* < 0.001, compared with HP-LCa^2+^ diet group.”

– Figure 6 – what tissue/specimen are panels D-F?

As noted in the figure legends, Figure 6D–F show kidney tissues. For clarity, we have added labels for Renal CYP27B1, VDR, and CYP24A1 in Figure 6D–F.

– Figure 7: I like the model, but key factors are missing? How is phytate metabolized and components absorbed? Is there a change in phytase from the gut microbiota? I may be biased, but it is my understanding that phytase is not present in the small intestine in mammals. Then, how is the digestibility change?

To address the reviewer’s questions, we analyzed the taxonomic composition of the gut microbiome at the phylum level using 16S rDNA quantitative PCR to quantify changes in intestinal microbial abundance from stool samples of rats fed control, HP-LCa^2+^, and HP-HCa^2+^ diets. We also performed immunohistochemical analysis for Ca^2+^ transporter (TRPV6), phosphate channel (NaPi-2b), and multiple inositol polyphosphate phosphatase 1 (MINPP1), a potential histidine phytase, using formalin-fixed intestinal tissue from rats fed control, HP-LCa^2+^, and HP-HCa^2+^ diets. To further address the reviewer’s questions regarding the changes in digestibility of phytate, we examined recent reports and several biological databases for information on intestinal or microbial phytases. Prior to our findings regarding phytate diets in rats, it was thought that the mammalian gut and intestinal microbiota did not possess phytases.

We have described our results in the Results section as follows:

“Changes in the microbiota of phytate-fed rats and the expression levels of intestinal multiple inositol polyphosphate phosphatase as a potential phytase

Given that phytate is digestible in the presence of low-Ca^2+^ diets but is indigestible in the presence of high Ca^2+^ diets in vivo, we examined whether phytate can be hydrolyzed by phytases from either intestinal microbiota or intestinal tissues. […] Taken together, these findings suggest crucial roles of gut bacterial and intestinal MINPP1 in the hydrolysis of dietary phytate, although further research is needed to determine whether the hydrolysis of phytate by MINPP1 is direct and/or indirect.”

Also, FGF23 is missing from the figure.

As shown in Figure 6—figure supplement 1D–E, we measured the serum levels of intact FGF23 and C-terminal FGF23 in rats fed control and high-phytate with low Ca^2+^ or high Ca^2+^ diets. We did not detect significant changes in the serum levels of intact FGF23 or C-terminal FGF23 between the three different dietary groups.

– Why is urinary pH changed?

It is not yet clear why a high-phytate/low-Ca^2+^ diet resulted in lower urinary pH compared to controls. A recent study showed that mean urinary pH was lower in kidney stone formers with hyperphosphaturia than in stone formers with normophosphaturia (Ha et al., 2010). As shown in Figure 5E and G, rats fed a high-phytate/low-Ca^2+^ diet showed a marked increase in the 24-hour urinary Pi/Cr ratio compared to controls, suggesting that rats fed high-phytate/low-Ca^2+^ diets excrete large amounts of phosphates in the urine. These results suggested that ~10 – 30-fold higher excretion of total urinary phosphate may slightly acidify the urinary pH in rats fed high-phytate/low-Ca^2+^ diets (pH 7.0 vs. pH 5.8). The decline of urinary pH in rats fed a high-phytate diet may represent significant renal acid loss in the presence of hyperphosphaturia.

– The main hypothesis is related to the digestion of phytate and absorption of minerals, but the intestine was not assessed. Did the authors collected small intestine? If so, I would recommend assessing changes in calcium and phosphorus transporters. Also, how exactly is phytate metabolized?

To address the reviewer’s questions, we checked the expression levels of calcium and phosphate transporters in the intestine by immunohistochemistry. There were no differences in the expression levels of the phosphate channel NaPi-2b and major Ca^2+^ channel TRPV6 in the intestines of rats fed control, high-phytate/low-Ca^2+^, or high-phytate/high-Ca^2+^ diets after 2 or 12 weeks.

We have briefly described the results in the revised manuscript as follows: “As phosphate and Ca^2+^ uptake occur along the gastrointestinal tract via the dedicated phosphate channel NaPi-2b and major Ca^2+^ channel TRPV6, we analyzed the expression levels of NaPi-2b and TRPV6 in the intestinal tissues by immunohistochemical staining. However, there were no differences in expression levels of intestinal NaPi-2b or TRPV6 between any of the different dietary groups (Figure 5—figure supplement 2A–D).”

Discussion– Members of the gut microbiota express phytase, there is no discussion on this and how the different types of diets may impact this.

We screened recent reports and several databases to address the reviewer’s questions. Based on our animal results and phylum analyses in rat feces, we have presented the results in the Discussion section as follows:

“Phytases can be classified into two major classes based on their biochemical and structural properties, i.e., the histidine acid phytases and the alkaline β-propeller phytases (Oh et al., 2004 and Sentz at al., 2014). […] Therefore, we propose that humans and other monogastric animals may lack β-propeller phytases that hydrolyze Ca^2+^-phytate in the gut, as high-phytate/high-Ca^2+^ diets could not be hydrolyzed (indigestible) in the guts of the rats and, eventually, most Ca^2+^-phytate was excreted in the feces.”

Supplementary figures– Figure 4—figure supplement 1: Is there a difference in FGF23?

Consistent with the results of rat studies, we did not detect any differences between three diets in human subjects. We emphasize that a high-phytate diet results in dysregulation of Ca^2+^ and phosphate homeostasis, thus contributing to the development of secondary hyperparathyroidism, independently of FGF23, which is known to suppress phosphate reabsorption by lowering active vitamin D levels.